# Hexokinase regulates Mondo-mediated longevity via the PPP and organellar dynamics

Raymond Laboy[1,2], Marjana Ndoci[1], Shamsh Tabrez Syed[1], Maximilian Vonolfen[1,2], Eugen Ballhysa[1,2], Tim Droth[1], Klara Schilling[1,2], Anna Loehrke[1], Ilian Atanassov[1], Adam Antebi[1,2]*

[1]Department of Molecular Genetics of Ageing, Max Planck Institute for Biology of Ageing, Cologne, Germany; [2]Cologne Excellence Cluster on Cellular Stress Responses in Aging-Associated Diseases (CECAD), University of Cologne, Cologne, Germany

## eLife Assessment

This **important** study utilizes the nematode *C. elegans* and mammalian cell culture to investigate the role of MML-1/Mondo in conserved regulation of metabolism and aging. The evidence supporting the conclusions is **convincing** and covers a range of areas including localization, upstream pathways, and conservation. The paper will be of interest to a broad range of biologists studying aging, metabolism, and transcriptional regulation.

*For correspondence:
antebi@age.mpg.de

**Abstract** The transcriptional complex Mondo/Max-like, MML-1/MXL-2, acts as a convergent transcriptional regulatory output of multiple longevity pathways in *Caenorhabditis elegans*. These transcription factors coordinate nutrient sensing with carbohydrate and lipid metabolism across the evolutionary spectrum. While most studies have focused on the downstream outputs, little is known about the upstream inputs that regulate these transcription factors in a live organism. Here, we found that knockdown of various glucose metabolic enzymes decreases MML-1 localization in the nucleus and identified two hexokinase isozymes, *hxk-1* and *hxk-2*, as the most vigorous regulators of MML-1 function. Upon hexokinase knockdown, MML-1 redistributes to mitochondria and lipid droplets (LDs), and concomitantly, transcriptional targets are downregulated and germline longevity is abolished. Further, we found that *hxk-1* regulates MML-1 through mitochondrial β-oxidation, while *hxk-2* regulates MML-1 by modulating the pentose phosphate pathway (PPP) and its coordinated association with LDs. Similarly, inhibition of the PPP rescues mammalian MondoA nuclear translocation and transcriptional function upon starvation. These studies reveal how metabolic signals and organellar communication regulate a key convergent metabolic transcription factor to promote longevity.

## Introduction

Aging is a complex, multifactorial process associated with the demise of organismal homeostasis and an increased risk of age-related diseases. One key aspect of aging is the dysregulation of gene expression and metabolism (*Fischer et al., 2022*). Many transcription factors that regulate metabolic pathways have been shown to control lifespan in the evolutionary spectrum (*Clancy et al., 2001*; *Green et al., 2022*; *Lapierre et al., 2013*; *Selman et al., 2008*; *Wang et al., 2008*). Changes in the activity of specific transcription factors have been shown to impact gene expression patterns and

alter metabolic processes, leading to a decline in cellular function and an increased risk of age-related diseases (*Benhamed et al., 2012*; *Carroll et al., 2015*; *Denechaud et al., 2008*; *Du and Zheng, 2021*; *Tong et al., 2009*). Thus, understanding the interplay between metabolism and transcriptional regulation is crucial for gaining insight into the mechanisms underlying aging and developing novel strategies for promoting healthy aging and reducing the risk of age-related diseases.

MondoA and the carbohydrate response element binding protein (ChREBP) are paralogous transcription factors that work in heterodimeric complexes with Max-like protein X (Mlx) and have been identified as key regulators of glucose metabolism and energy homeostasis. These transcription factors recognize and bind hexanucleotide tandem sequences (CANNTG) in the promoter regions called Enhancer box (E-box) sequences (*Massari and Murre, 2000*) of various genes involved in glucose uptake, utilization, and storage (*Billin et al., 2000*; *Cha-Molstad et al., 2009*; *Sans et al., 2006*; *Stoltzman et al., 2008*; *Yamashita et al., 2001*). In addition to their role in glucose metabolism, MondoA/ChREBP has also been shown to regulate lipid metabolism and cellular stress responses (*Iizuka et al., 2006*; *Ishii et al., 2004*; *Ma et al., 2006*). Studies have suggested that changes in MondoA/ChREBP activity with age may contribute to age-related metabolic dysfunction, leading to a decline in glucose tolerance and increased risk of type 2 diabetes and obesity (*Ghasemi et al., 2015*; *Iizuka et al., 2006*; *Richards et al., 2017*; *Yamamoto-Imoto et al., 2022*).

In *Caenorhabditis elegans,* the transcriptional complex formed by MML-1 (MondoA/ChREBP homolog) and MXL-2 (Mlx homolog) is part of an integrated helix-loop-helix network required for lifespan extension in diverse longevity pathways (*Johnson et al., 2014*; *Nakamura et al., 2016*). However, how these transcription factors integrate multiple upstream signals to regulate lifespan remains unknown. This work describes how metabolic inputs regulate MML-1 subcellular localization and function. We found that HXK-1 and HXK-2 regulate MML-1 cellular distribution and are required for germline longevity by regulating mitochondrial β-oxidation and the pentose phosphate pathway (PPP), respectively. Moreover, inhibition of MML-1 nuclear translocation results in the relocalization of this transcription factor to mitochondria associated with lipid droplets (LDs). This study provides key insights into how glucose and lipid metabolism directly impact transcriptional regulation and aging.

## Results

### Hexokinases are required for MML-1 subcellular localization and longevity

MondoA reportedly responds to glucose 6-phosphate (G6P) and other hexose phosphate sugars in cultured cells by translocating to the nucleus and regulating the expression of its downstream targets (*Stoltzman et al., 2008*; *Stoltzman et al., 2011*). Nevertheless, little is known about MondoA regulation in an organism-wide setting, and most subsequent work has focused on the molecular mechanisms downstream of this transcription factor. Hence, we decided to investigate how MML-1 is regulated by upstream glucose metabolism in *C. elegans*. To address this, we performed a targeted RNA interference (RNAi)-based screen on genes encoding enzymes involved in glucose metabolism. Overall, we found a preponderance of decreased intestinal MML-1 nuclear localization when knocking down such genes (*Figure 1A and B*, *Supplementary file 1A*). Knockdown of most of the enzymes involved in glycolysis, as well as the pyruvate dehydrogenase complex, which links glycolysis to the tricarboxylic acid (TCA) cycle via acetyl-CoA, decreased MML-1 nuclear accumulation (*Supplementary file 1*). Interestingly, whereas most enzymes in the TCA cycle also decreased nuclear MML-1, inhibition of the oxoglutarate dehydrogenase complex (OGDC) subunits *ogdh-1* and *dld-1* and the succinyl-CoA synthase (SCS) *suca-1* and *sucg-1* subunits increased MML-1 nuclear localization. We also observed that knockdown of enzymes involved in the oxidative and nonoxidative branches of the PPP, phosphogluconate dehydrogenase (PGDH) *T25B9.9* and transketolase (TKT) *tkt-1* genes, respectively, triggered a strong increase in MML-1 nuclear accumulation (*Figure 1A and B*). These data suggest that overall changes in glucose metabolism regulate MML-1 and that this transcription factor may sense the divergence of G6P to different branches of this network.

Our screen revealed that hexokinase knockdown had the most potent effect in decreasing MML-1 nuclear localization. This enzyme is involved in the first step of glucose metabolism, phosphorylating glucose to G6P and is a critical mediator of cellular metabolism. Among the three *C. elegans* hexokinase genes, *hxk-1* and *hxk-2* more strongly affected MML-1 nuclear localization in two independent

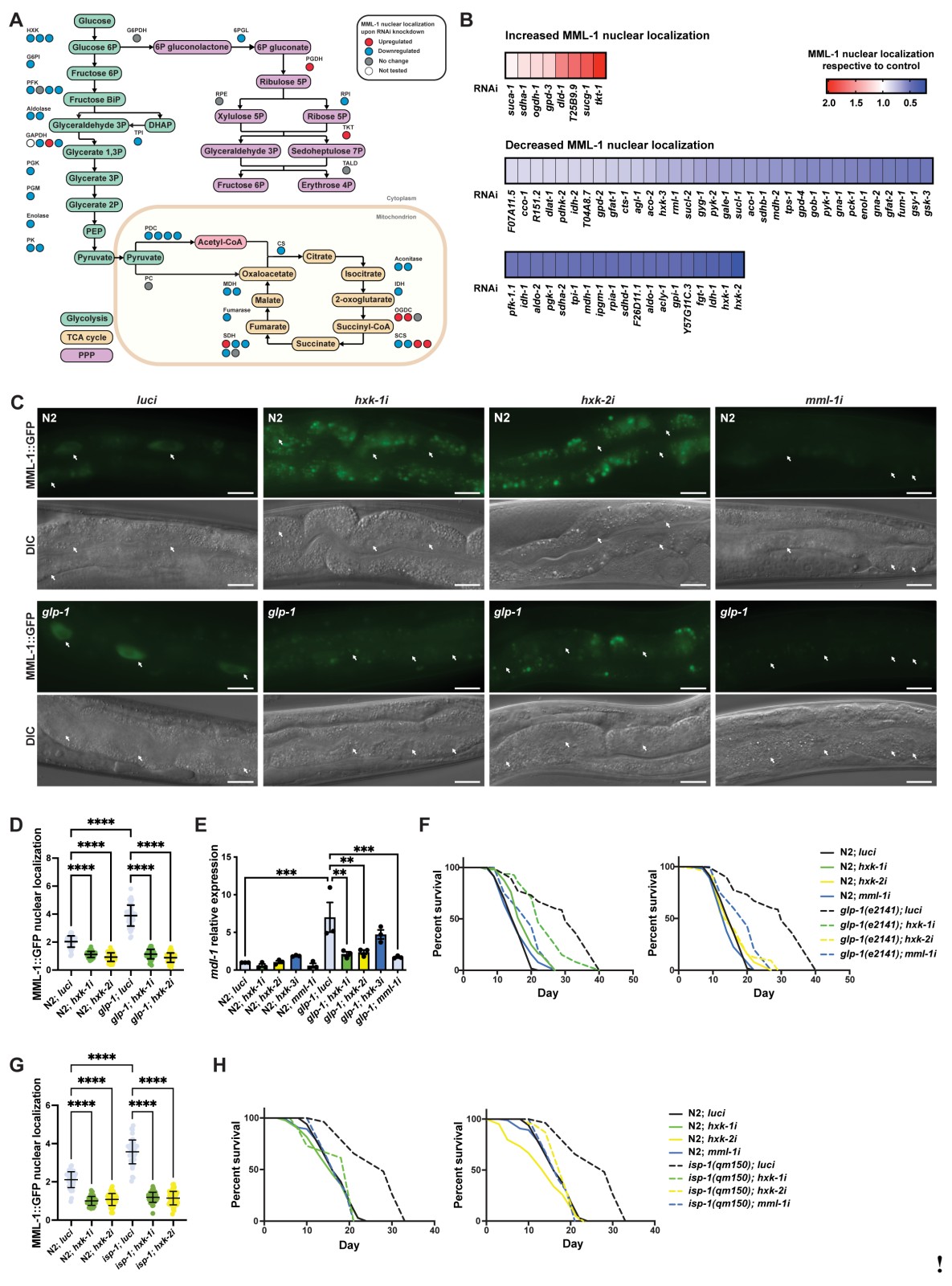

**Figure 1.** Hexokinases are required for germline longevity by regulating MML-1 localization and function. (**A**) Screening for MML-1 regulators upon knockdown of glucose metabolism enzymes. For clarity, we only mapped the candidate genes involved in glycolysis, the TCA cycle, and PPP. See *Supplementary file 1A* for a complete list of genes tested. MML-1::GFP (dhIs989) nuclear localization was measured in intestinal cells of day 1 adult worms grown in the RNA interference (RNAi) egg-on. Each circle represents one gene encoding an isozyme or subunit of the complex. Circles are

*Figure 1 continued on next page*

*Figure 1 continued*

color-coded to indicate the increase (red) or decrease (blue) of MML-1::GFP nuclear localization upon knockdown of the enzymes compared to internal *luci* controls. DHAP, dihydroxyacetone phosphate; PEP, phosphoenolpyruvate; TCA cycle, tricarboxylic acid cycle; PPP, pentose phosphate pathway. (**B**) Quantification of MML-1 nuclear localization upon knockdown of glucose metabolism genes. Only genes that statistically affected MML-1 nuclear localization compared to the internal *luci* control are depicted (p<0.05; N≥2). (**C, D**) Representative images (**C**) and quantification (**D**) of MML-1 nuclear localization in wild type and the germlineless *glp-1(e2141)* longevity background upon knockdown of *luci* (control), *hxk-1i*, *hxk-2i*, and *mml-1i*. MML-1::GFP was quantified in nuclei from intestinal cells (white arrows) in day 1 adult worms (N=3). Scale bars, 10 μm. (**E**) Relative mRNA levels of MML-1 downstream target *mdl-1* in young adult worms measured by qPCR (N=3). (**F**) Lifespans of wild type and *glp-1(e2141)* upon *hxk-1* and *hxk-2* knockdown. Lifespans for *hxk-1* and *hxk-2* knockdown were performed in the same experiment and plotted separately for clarity; therefore, control lifespans are shared between plots. (**G**) Quantification of MML-1 nuclear localization in wild-type and *isp-1(qm150)* mutants upon *luci*, *hxk-1i*, and *hxk-2i* in day 1 adult worms (N=3). (**H**) Lifespan of wild type and *isp-1(qm150)* upon *hxk-1* and *hxk-2* knockdown (N=2). Statistical significance was calculated with t-test in (**A, B**), one-way ANOVA in (**D, G**), two-tailed t-test in (**E**), and Log-rank (Mantel-Cox) test in (**F, H**). (**D, G**) bars represent mean ± SD, and (**E**) bars represent mean ± SEM.

The online version of this article includes the following source data and figure supplement(s) for figure 1:

**Source data 1.** Raw data from *Figure 1*.

**Figure supplement 1.** Hexokinases regulate MML-1 nuclear localization and transcriptional function.

**Figure supplement 1—source data 1.** Original western blots from *Figure 1—figure supplement 1*.

**Figure supplement 1—source data 2.** Original western blots from *Figure 1—figure supplement 1*, labeled.

**Figure supplement 1—source data 3.** Raw data from *Figure 1—figure supplement 1*.

MML-1::GFP reporter strains (*Figure 1B*, *Figure 1—figure supplement 1A*), while *hxk-3* had just a small effect on MML-1 nuclear localization, probably due to its low expression in adult worms (*Hutter and Suh, 2016*). To test whether the phosphotransferase activity of hexokinase is responsible for regulating MML-1 nuclear localization, we used two drugs to inhibit hexokinases pharmacologically. 3-Bromopyruvate (3-BrP) is an alkylating agent and HKII inhibitor, and 2-deoxy-glucose (2-DG) is an analog of glucose phosphorylated to 2-DG6P, which cannot be further metabolized, thus inhibiting hexokinase function. Supplementation of both drugs significantly decreased MML-1 nuclear localization compared to control (*Figure 1—figure supplement 1B and C*), indicating that hexokinase activity drives MML-1 nuclear translocation. We also tested the specificity of RNAi knockdown by using strains where the three endogenous hexokinase loci were tagged with mKate2. We used these strains to evaluate the protein levels by western blot (WB) and found that each RNAi was specific to each isozyme (*Figure 1—figure supplement 1D and E*). As glycolysis is essential during development, we measured the size (area and length) of worms fed with hexokinase RNAi egg-on. *hxk-1i* and *hxk-2i* displayed no significant change in worm growth, while *hxk-3i*-treated animals were somewhat smaller (*Figure 1—figure supplement 1F and G*). To determine if our observations were due to a change in food intake, we counted pharyngeal pumping rate but found no significant difference compared to the control (*Figure 1—figure supplement 1H*), nor did we see an effect on worm motility compared to control (*Figure 1—figure supplement 1I*). Moreover, hexokinase knockdown did not affect the steady-state levels of *mml-1* mRNA (*Figure 1—figure supplement 1J*). Taken together, these data indicate no RNAi cross-reactivity among the hexokinase isozymes and that the effects on MML-1 nuclear localization are not due to a decrease in food consumption or developmental phenotype.

MML-1 works in a transcriptional complex with MXL-2, and previous studies have demonstrated that MML-1 and MXL-2 are required for lifespan extension in multiple longevity pathways, including germline-less *glp-1(e2141)* mutant (*Nakamura et al., 2016*). Hence, we tested whether hexokinases influenced MML-1 subcellular localization in this longevity model. As expected, MML-1 nuclear localization was increased in *glp-1(e2141)* as reported previously (*Nakamura et al., 2016*), and upon *hxk-1* and *hxk-2* knockdown, MML-1 nuclear localization was abolished in *glp-1(e2141)* mutants (*Figure 1C and D*). Changes in MML-1 nuclear localization would be expected to modulate its transcriptional activity. To test this idea, we next analyzed the expression of downstream targets reportedly linked to gonadal longevity (*Nakamura et al., 2016*). We observed decreased expression of *mdl-1*, *lgg-2*, *swt-1*, and *fat-5* under *hxk-1i* and *hxk-2i*, but not under *hxk-3i* (*Figure 1E*, *Figure 1—figure supplement 1K*), consistent with decreased MML-1 transcriptional activity. Since hexokinase knockdown decreases MML-1 nuclear localization in the *glp-1(e2141)* longevity model, we reasoned that the lifespan might also be abolished. Accordingly, *hxk-1i* and *hxk-2i* significantly decreased *glp-1(e2141)* but did not

affect wild-type lifespan (*Figure 1F*). These data indicate that hexokinases are upstream regulators of MML-1 nuclear localization and transcriptional function in germline-mediated longevity.

Next, we tested whether the effect of hexokinase knockdown on MML-1 nuclear localization and longevity was shared by other longevity pathways. On the one hand, we observed that in *isp-1(qm150)* mutants, a model for mitochondrial longevity, *hxk-1* and *hxk-2* knockdown decreased MML-1 nuclear localization and abolished lifespan extension (*Figure 1G and H*). On the other hand, while *mml-1* was required for the insulin signaling *daf-2(e1370)* mutant longevity, *hxk-1* and *hxk-2* knockdown did not alter MML-1 nuclear localization nor suppress the lifespan in this background (*Figure 1—figure supplement 1L–M*). Further, we found that MML-1 nuclear localization was increased in *raga-1(ok701)* mutants, a model for mTORC1 longevity, and *mml-1* was required for lifespan extension (*Figure 1— figure supplement 1N and O*). However, *hxk-1* and *hxk-2* knockdown did not suppress *raga-1(ok701)* longevity (*Figure 1—figure supplement 1P*). Collectively, these data suggest that while MML-1 acts as a convergent factor in these four longevity pathways, it only requires HXK-1 and HXK-2 for germline and mitochondrial longevity, but not for insulin and mTOR longevity.

## *C. elegans* hexokinases are differentially expressed

Mammals harbor four hexokinases that differ in tissue expression, subcellular localization, and affinity to glucose and G6P. However, knowledge of the worm isozymes is limited. We used the hexokinase tagged with mKate2 strains to investigate the tissue-specific expression and subcellular localization of the different isozymes by confocal microscopy. We found that HXK-1 was expressed in the cytosol of neurons, pharynx, gonadal sheath, and coelomocytes (*Figure 2A*). HXK-2 had a reticular expression pattern in neurons, muscle, intestine, and hypodermis (*Figure 2B*). HXK-3 signal was diffused and highly expressed in the pharynx, muscle, hypodermis, and intestine (*Figure 2—figure supplement 1A*).

Human HKI and HKII have been shown to bind reversibly to the mitochondria (*Sui and Wilson, 1997*). Binding depends on an intrinsic hydrophobic N-terminal sequence critical for inserting into the outer mitochondrial membrane (*Pastorino et al., 2002*). Due to the reticular expression pattern, HXK-2 was a plausible candidate to be mitochondrially localized. To corroborate this idea, we crossed the HXK-2::mKate2 strain with a mitochondrial reporter that expresses GFP with a mitochondrial targeting sequence under the body wall muscle promoter *myo-3*. Indeed, we observed that full-length HXK-2 strongly co-localized with mitochondria (*Figure 2C*). We also generated an HXK-2 mutant that lacks the first 10 amino acids after the initiator methionine (d10 HXK-2) and found that mitochondrial localization was abolished entirely (*Figure 2C*). In summary, we found that hexokinases are differentially expressed in *C. elegans* and identified HXK-2 as the worm mitochondrial hexokinase.

## Hexokinase knockdown increases MML-1 mitochondrial localization

Since we observed that *mml-1* mRNA levels were unaffected by *hxk* knockdown and saw a decrease in MML-1 nuclear localization upon hexokinase knockdown, we wondered whether MML-1 might re-localize to another subcellular compartment. First, we crossed the MML-1::GFP reporter strain with two compartmental markers and examined co-localization by confocal microscopy. We looked at the lysosome/endosome reporter strain LMP-1::mKate2 but saw no evident co-localization in intestinal cells upon *hxk-2* knockdown or control conditions (*Figure 2—figure supplement 1B and C*). We had previously shown that MML-1 localizes with mitochondria in the hypodermis (*Nakamura et al., 2016*); however, mitochondrial subcellular localization in other tissues had not been examined. Hence, we crossed the MML-1::GFP strain with the mitochondrial reporter TOMM-20::mCherry under the intestinal promoter *ges-1* and confirmed that MML-1 localized with mitochondria in the intestine (*Figure 2D and E*). Under normal conditions, around 60–80% of the total MML-1 was nuclear, and 15% co-localized with mitochondria. Interestingly, MML-1 mitochondrial localization was increased to ~62% of total MML-1 upon *hxk-2i* (*Figure 2F*). Taken together, our data indicate that upon *hxk-2* knockdown, MML-1 is excluded from the nucleus and re-localizes to a subset population of mitochondria.

## Inhibition of mitochondrial β-oxidation rescues MML-1 nuclear localization under hxk-1i

Many genes that regulate MML-1 localization are involved in mitochondrial metabolism, such as OGDC and SCS, and inhibiting these complexes could activate stress response pathways. Hence,

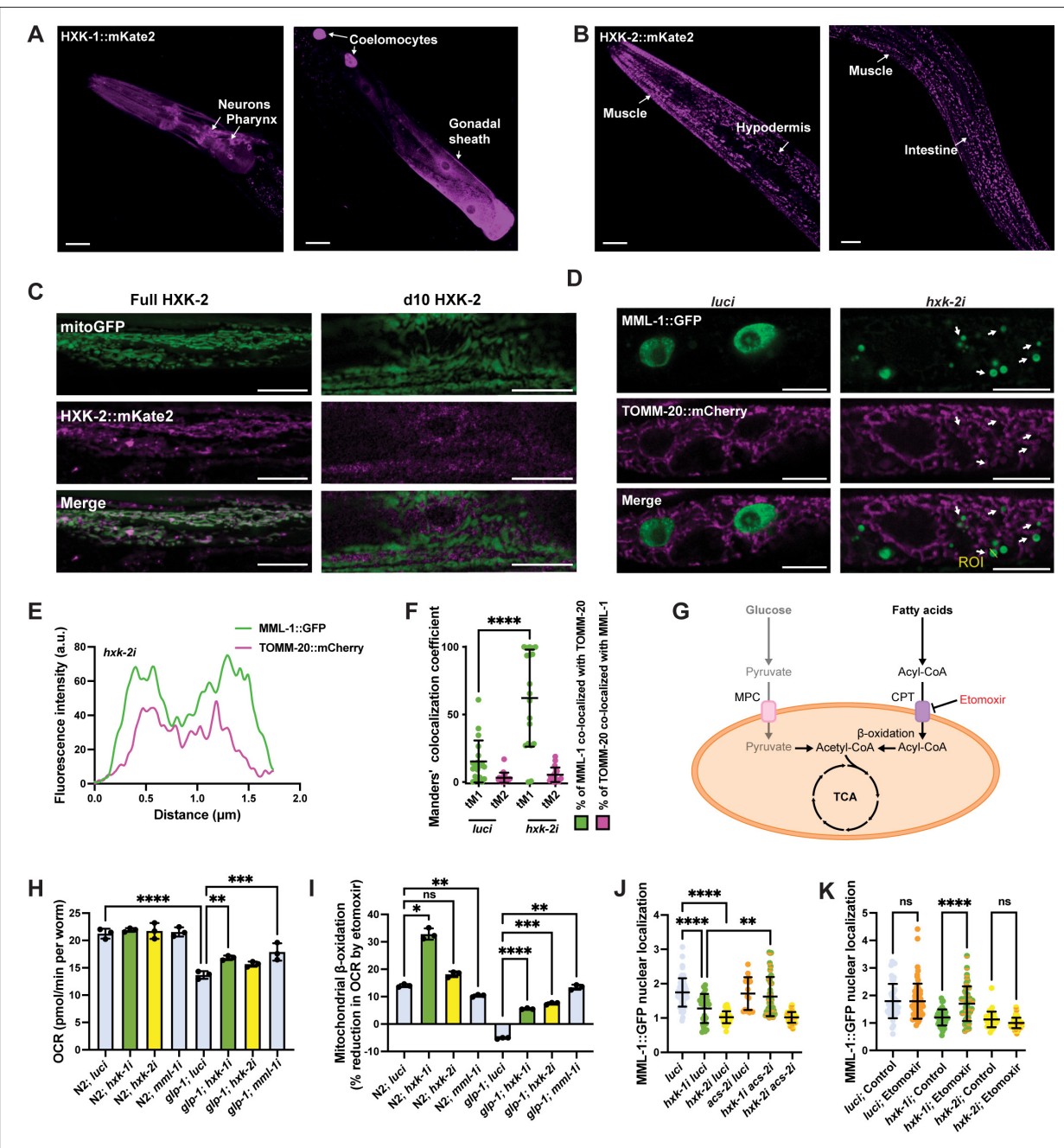

**Figure 2.** Enhanced mitochondrial β-oxidation suppresses MML-1 nuclear localization upon *hxk-1* knockdown. (**A, B**) Tissue expression of HXK-1::mKate2 (**A**) and HXK-2::mKate2 (**B**) isozymes. The primary tissues are identified with arrows. Scale bars, 20 μm. (**C**) Co-localization of mitochondrial reporter *myo-3p*::mitoGFP with full-length HXK-2::mKate2 and d10 HXK-2::mKate2 lacking the first 10 residues of the N-terminus. Scale bars, 10 μm. (**D**) Representative confocal images of day 1 adult worms expressing MML-1::GFP and the mitochondrial reporter *ges-1p*::TOMM-20::mCherry under *luci* and *hxk-2i*. Scale bars, 10 μm. (**E**) Intensity plot of the region of interest (ROI) from *hxk-2i* (**D**). (**F**) Co-localization of MML-1::GFP and TOMM-20::mCherry in *luci* and *hxk-2i*. The co-localization analysis was calculated with the thresholded Manders' co-localization coefficients tM1 (GFP channel) and tM2 (mCherry channel) (N=3). (**G**) When glycolysis is reduced, cells rely on fatty acids for energy production. Free fatty acids are activated to acyl-CoA and transported to the mitochondria through the CPT in the mitochondrial outer membrane. The acyl-CoA is then metabolized to generate acetyl-CoA that can enter the tricarboxylic acid (TCA) cycle to produce energy. MPC, mitochondrial pyruvate carrier; CPT, carnitine palmitoyltransferase. (**H, I**) Oxygen consumption rate (OCR) under basal conditions (**H**) and after the addition of the mitochondrial β-oxidation inhibitor etomoxir (**I**). Mitochondrial β-oxidation is calculated as the percentage of decreased OCR after injecting etomoxir (N=3). (**J**) MML-1 nuclear localization of day 1 adult worms grown with double knockdown of *hxk-1* or *hxk-2* combined *acs-2i* (N=3). (**K**) MML-1 nuclear localization of day 1 adult worms grown upon *hxk-1* and *hxk-2* knockdown with supplementation of etomoxir 100 μM egg-on (N=3). Statistical significance was calculated with t-test in (**F**), and one-way ANOVA in (**H–K**). (**F, J–K**) bars represent mean ± SD, (**H, I**) bars represent mean ± SEM.

*Figure 2 continued on next page*

*Figure 2 continued*

The online version of this article includes the following source data and figure supplement(s) for figure 2:

**Source data 1.** Raw data from *Figure 2*.

**Figure supplement 1.** MML-1 regulation by glucose metabolism is independent of mitochondrial and oxidative stress response.

**Figure supplement 1—source data 1.** Raw data from *Figure 2—figure supplement 1*.

we assessed whether MML-1 nuclear localization was associated with the activation of mitochondrial stress response and oxidative stress. As expected, we observed increased expression of *hsp-6::GFP*, a reporter for mitochondrial stress, upon knockdown of mitochondrial components *ogdh-1*, *sucl-1*, and *sucl-2*, as well as in control worms treated with antimycin A (AA), a complex III inhibitor (**Figure 2— figure supplement 1D**). However, inducing mitochondrial stress with AA did not affect MML-1 nuclear localization under control conditions and *hxk-2* knockdown (**Figure 2—figure supplement 1E**). Next, we sought to test whether an increase in oxidative stress affects MML-1 nuclear localization. We exposed worms to the ROS-generating compound paraquat and observed induction of the oxidative reporter *gst-4p*::GFP (**Figure 2—figure supplement 1F**); however, we did not see any effect on MML-1 nuclear localization at the concentrations tested (**Figure 2—figure supplement 1G**). Taken together, these data indicate that MML-1 regulation by glucose metabolism genes is independent of mitochondrial and oxidative stress.

Hexokinase downregulation might be expected to cause a reduction in the reliance on glycolysis and increase fatty acid metabolism as fuel for cellular maintenance (**Figure 2G**). Mitochondrial β-oxidation is an important pathway that generates acetyl-CoA for the TCA cycle through the catabolism of fatty acids to produce reductants and energy. Hence, we sought to investigate whether decreasing fatty acid oxidation could affect MML-1 nuclear localization upon hexokinase knockdown. First, we measured the neutral lipid content by feeding the worms with the labeled fatty acid C1-BODIPY-C12 and found increased total lipid storage under hexokinase knockdown in the wild-type and *glp-1(e2141)* backgrounds (**Figure 2—figure supplement 1H**). Next, we measured the oxygen consumption rate (OCR) upon hexokinase knockdown under basal conditions and with etomoxir supplementation, a mitochondrial β-oxidation inhibitor that acts on the carnitine palmitoyltransferase (CPT). Inhibition of the CPT decreases acyl-CoA import to mitochondria, and changes in OCR are a correlative measure of mitochondrial β-oxidation of fatty acids (**Ramachandran et al., 2019**). We found no difference in basal OCR under *hxk-1*, *hxk-2*, and *mml-1* knockdown in the wild-type background (**Figure 2H**). *glp-1(e2141)* animals showed decreased basal OCR compared to wild type, whereas *hxk-1* and *mml-1* knockdown significantly increased the basal OCR compared to *glp-1(e2141)* (**Figure 2H**). Interestingly, we saw a greater percentage of OCR reduction upon adding etomoxir under *hxk-1* knockdown in wild-type and *hxk-1*, *hxk-2*, and *mml-1* knockdown in *glp-1(e2141)* (**Figure 2I**), indicating higher mitochondrial β-oxidation levels. We confirmed this result using the transcriptional reporter *acs-2p*::GFP, a mitochondrial acyl-CoA synthase that is induced and required for fatty acid β-oxidation in *C. elegans* (**Van Gilst et al., 2005**), and found increased expression upon *hxk-1* and *hxk-2* knockdown (**Figure 2—figure supplement 1I**).

We next wondered whether this increase in fatty acid oxidation was responsible for decreased MML-1 nuclear localization under hexokinase knockdown. First, we performed a double knockdown of *hxk-1* or *hxk-2* combined with *acs-2*. MML-1 nuclear localization was rescued upon *acs-2i* under *hxk-1i* (**Figure 2J**). Interestingly, *acs-2i* did not rescue MML-1 nuclear localization under *hxk-2i*. We also pharmacologically inhibited fatty acid oxidation by supplementing etomoxir and again saw the rescue of MML-1 nuclear localization under *hxk-1i,* but no effect upon *hxk-2i* (**Figure 2K**). These data suggest that HXK-1 normally promotes MML-1 nuclear localization by decreasing mitochondrial β-oxidation and reveal that the two hexokinases might regulate MML-1 nuclear localization through somewhat independent pathways.

## hxk-2 regulates MML-1 function through increased PPP

G6P can be diverted as a substrate for metabolic pathways other than glycolysis. Thus, we wanted to test whether MML-1 could sense other branches of glucose metabolism. From our RNAi screen, we observed increased MML-1 nuclear localization upon knockdown of two enzymes from the PPP, *T25B9.9* and *tkt-1* (**Figure 1A and B**). The PPP is a cytosolic metabolic pathway involved in the

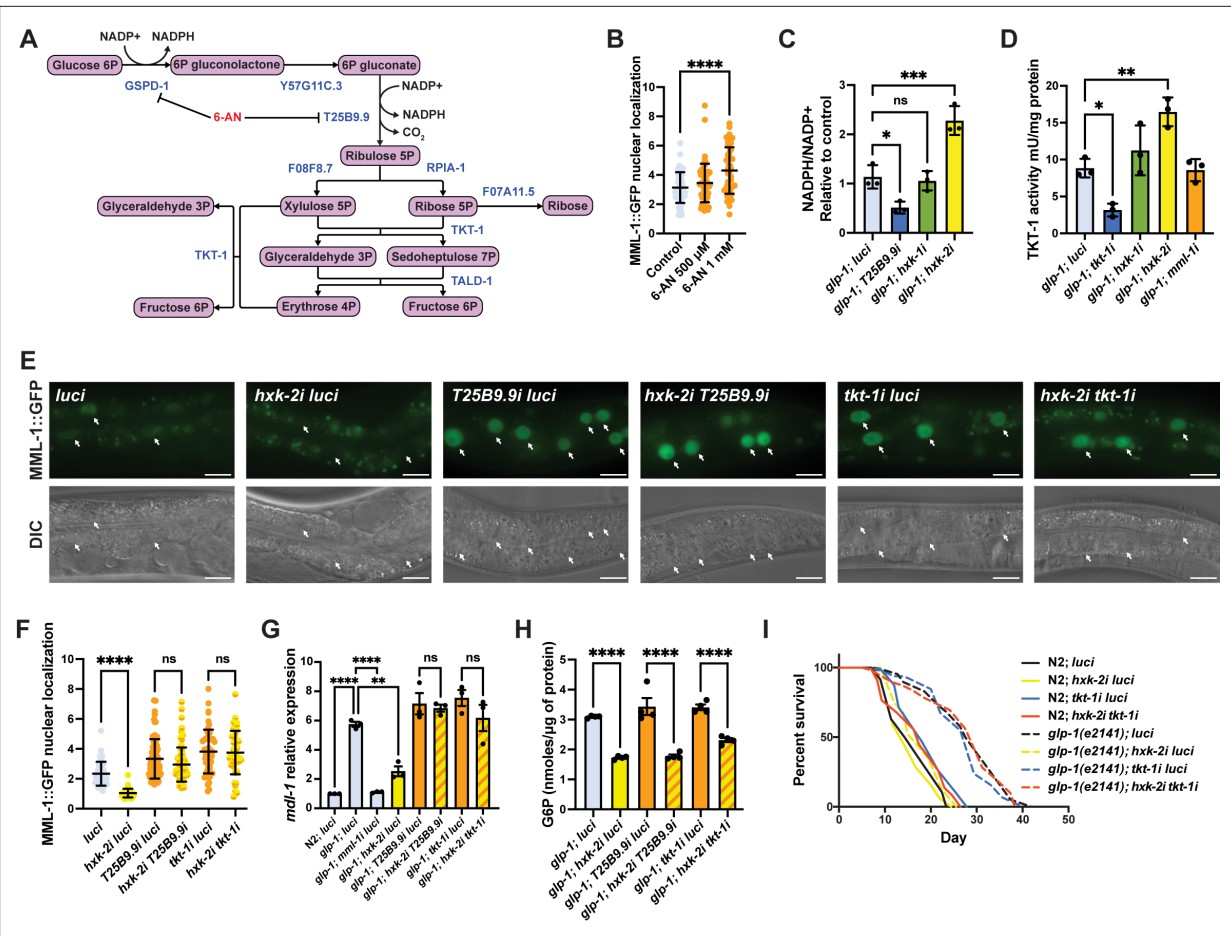

**Figure 3.** HXK-2 regulates MML-1 localization and function through inhibition of the pentose phosphate pathway (PPP). (**A**) The PPP is a cytosolic pathway that metabolizes glucose 6-phosphate (G6P) to produce NADPH and ribose-5P, and interconverts 3–7 carbon sugars. 6-Aminonicotinamide (6-AN) is a PPP inhibitor that acts on the NADPH-producing enzymes GSPD-1 and T25B9.9. (**B**) Quantification of MML-1 nuclear localization of day 1 adult worms supplemented with 500 µM and 1 mM of 6-AN egg-on (N=3). (**C**) Quantification of NADP+ and NADPH levels in day 1 adult *glp-1(e2141)* worms grown under hexokinase knockdown. *T25B9.9i* was used as a control. Metabolite levels were normalized to total protein concentration (N=3). (**D**) Transketolase enzymatic activity assay on day 1 adult *glp-1(e2141)* worms grown under hexokinase knockdown. *tkt-1i* was used as a control. Activity levels were normalized to total protein concentration (N=3). (**E, F**) Representative images (**E**) and quantification (**F**) of MML-1 nuclear localization of day 1 adult worms upon double knockdown of *hxk-2* with the PPP enzymes *T25B9.9* and *tkt-1* (N=3). Scale bars, 20 µm. (**G**) Relative mRNA levels of MML-1 downstream target *mdl-1* in young adult worms upon double knockdown of *hxk-2* in combination with enzymes of the PPP measured by qPCR (N=3). (**H**) Quantification of G6P levels relative to total protein measured on day 1 adult *glp-1(e2141)* upon double knockdown of *hxk-2* in combination with enzymes of the PPP (N=4). (**I**) Lifespan analysis of *glp-1(e2141)* with *hxk-2i* and *tkt-1i* single and double knockdowns. Lifespan was performed in the same experiment as in *Figure 3—figure supplement 1J*; therefore, control lifespans are shared between plots (N=3). Statistical significance was calculated with a one-way ANOVA test in (**B–D, F–H**), and Log-rank (Mantel-Cox) test in (**I**). (**B, F**) bars represent mean ± SD, (**C, D, G, H**) bars represent mean ± SEM.

The online version of this article includes the following source data and figure supplement(s) for figure 3:

**Source data 1.** Raw data from *Figure 3*.

**Figure supplement 1.** HXK-1 and HXK-2 regulate MML-1 through parallel or independent pathways.

**Figure supplement 1—source data 1.** Raw data from *Figure 3—figure supplement 1*.

interconversion of 3- to 7-carbon sugars that generate precursors for the biosynthesis of lipids, amino acids, and nucleotides (*Figure 3A*). It also maintains redox homeostasis through NADPH production. To further address these findings, we tested the effect of pharmacological inhibition of the PPP on MML-1 nuclear localization using 6-aminonicotinamide (6-AN), which is a competitive inhibitor of the G6P dehydrogenase and 6-phosphogluconate dehydrogenase (6PGD). These two enzymes generate NADPH in the PPP. In agreement with our previous results, we observed increased MML-1 nuclear localization after supplementation with 1 mM of 6-AN (*Figure 3B*). Moreover, we found that *hxk-2i*,

but not *hxk-1i*, significantly increased the NADPH levels (*Figure 3C*). Additionally, we found that *hxk-2i* had increased TKT-1 enzymatic activity compared to the control (*Figure 3D*). These data suggest that inhibition of the mitochondrial hexokinase enhances the PPP.

Next, we sought to test the epistatic interaction of hexokinase and PPP knockdown on MML-1 nuclear localization. For these experiments, we performed double RNAi of *hxk-1* and *hxk-2* in combination with the 6PGD *T25B9.9* and transketolase *tkt-1*. Interestingly, knocking down these enzymes rescued MML-1 nuclear localization under *hxk-2i* (*Figure 3E and F*) but did not rescue the localization under *hxk-1i* (*Figure 3—figure supplement 1A*). To evaluate MML-1 function, we measured its transcriptional readouts and found that knocking down the PPP under *hxk-2i* rescued the expression of *lgg-2*, *mdl-1*, *swt-1*, and *fat-5* (*Figure 3G*, *Figure 3—figure supplement 1B–D*); however, it had no effect under *hxk-1i* (*Figure 3—figure supplement 1E–H*). Moreover, we observed lower levels of G6P upon double knockdown of hexokinases with *T25B9.9* and *tkt-1* compared to control (*Figure 3H*, *Figure 3—figure supplement 1I*). Taken together, these data suggest that the PPP activity is increased upon knockdown of the mitochondrial hexokinase HXK-2, and inhibiting the PPP is sufficient to rescue MML-1 function independent of G6P levels.

Finally, we asked whether reducing the PPP could impact *glp-1(e2141)* lifespan upon *hxk-2* knockdown. Inhibition of the PPP has been reported to extend the lifespan in *C. elegans* by activating the mitochondrial UPR^mt (*Bennett et al., 2017*). As reported previously, we found a significant lifespan extension upon *tkt-1* knockdown in wild type (*Figure 3I*). Moreover, knockdown of the PPP in the *glp-1(e2141)* background had no additive effect on longevity, suggesting that *glp-1* and *tkt-1i* may share common mechanisms in extending lifespan. Consistent with our data, knockdown of *T25B9.9* and *tkt-1* rescued *glp-1(e2141)* longevity upon *hxk-2i* (*Figure 3I*, *Figure 3—figure supplement 1J*). Collectively, these data indicate that an increase in the PPP under *hxk-2* knockdown suppresses MML-1 function, and reducing PPP restores longer life, placing PPP epistatically downstream of *hxk-2*.

## MML-1 interactome uncovers potential regulators of its function and localization

Given the mitochondrial localization of MML-1 and possibly other organelles, we decided to identify potential MML-1 binding partners to better understand its subcellular distribution. We used CRISPR/Cas9 to tag the endogenous MML-1 locus with a 3xFLAG tag. MML-1::3xFLAG was immunoprecipitated from whole-worm lysates, and potential interactors were identified by mass spectrometry. We detected ca. 1600–2000 different proteins from three biological replicates and observed that the biological replicates clustered together (*Figure 4—figure supplement 1A and B*). We detected several proteins in the immunoprecipitation (IP), including MXL-2, an established MML-1 binding partner (*Pickett et al., 2007*; *Figure 4A*). Analysis of the top 50 co-enriched candidates in MML-1 IPs revealed enrichment for nuclear and mitochondrial proteins (*Figure 4—figure supplement 1C*). We also found multiple enriched proteins from different organelles, including LDs.

Mitochondria and LD dynamically interact in highly active metabolic tissues, and it has been shown that mitochondria associated with LD have specific metabolic behavior compared to cytosolic mitochondria (e.g. different dynamics, motility, and capacity to burn carbohydrates and lipids) (*Gordaliza-Alaguero et al., 2019*; *Rambold et al., 2015*). LDs are organelles involved in multiple roles serving as nutrient reservoirs and participating in signaling pathways, and their biogenesis or degradation is tightly coupled to carbohydrate metabolism (*Figure 4B*). Interestingly, PLIN-1, one of the most abundant LD proteins, was co-enriched in MML-1 IP (*Figure 4A*). Hence, we wanted to test whether MML-1 localized to these organelles. First, we used LipiBlue, a dye to specifically visualize LD in vivo (*Tatenaka et al., 2019*), and found that all LipiBlue-stained structures were positive with the LD-resident proteins DHS-3::GFP and PLIN-1::mCherry (*Figure 4—figure supplement 1D–F*), indicating that this dye could be used as a reliable method for visualization of LD in *C. elegans*. Next, we examined LipiBlue-stained animals co-expressing MML-1::GFP and TOMM-20::mCherry and saw that MML-1 localized with LipiBlue-positive structures (*Figure 4C*). Interestingly, MML-1 appeared localized to mitochondria associated with LD (*Figure 4C and D*). Moreover, this co-localization was significantly increased upon *hxk-2* knockdown (*Figure 4E*).

Inhibition of transaldolase and enzymes from the PPP has been shown to activate a fasting-like response by increasing lipolysis and enhancing the breakdown of LD (*Bennett et al., 2017*), including higher expression of the adipose triglyceride lipase ATGL-1 (*Figure 4—figure supplement 1G*).

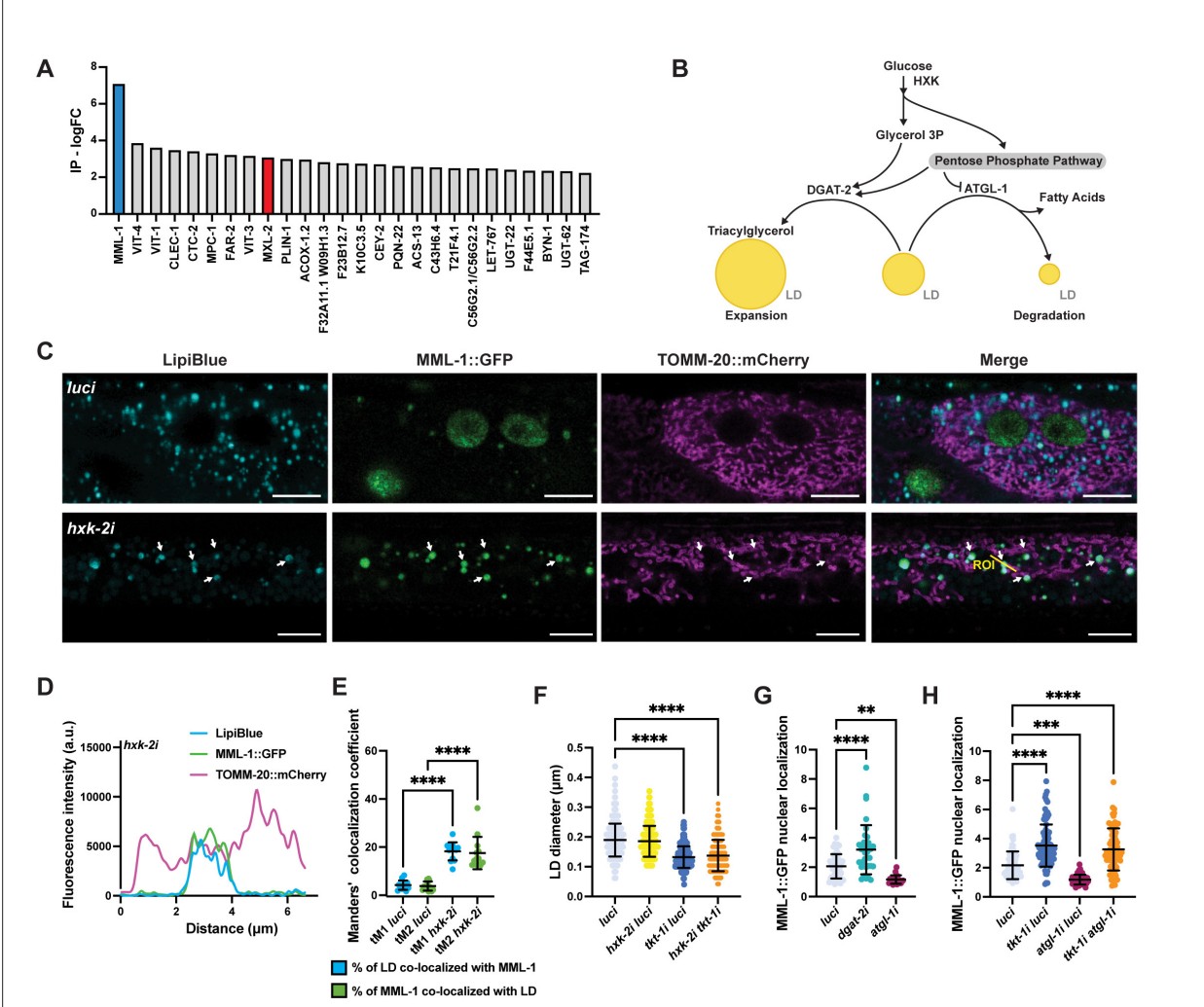

**Figure 4.** *hxk-2* knockdown increases MML-1 co-localization with lipid droplet (LD). (**A**) Anti-FLAG immunoprecipitation (IP) from whole-worm lysates of animals expressing MML-1 endogenously tagged with 3xFLAG in wild-type background. IP was analyzed by mass spectrometry (LCMS/MS). The 25 top candidates co-enriched with MML-1 compared to non-transgenic wild-type control are shown as log FC, including MML-1 (blue) and its established binding partner MXL-2 (red). (**B**) Crosstalk between glycolysis, pentose phosphate pathway (PPP), and LD metabolism. On the one hand, LD expansion is favored by increased fatty acid synthesis and subsequent formation of triacylglycerol (TAG) by the diacylglycerol acetyltransferase DGAT-2. On the other hand, LD consumption releases free fatty acids from TAG by the rate-limiting lipase ATGL-1. (**C**) Confocal microscopy of day 1 adult worms expressing MML-1::GFP and the mitochondrial reporter TOMM-20::mCherry stained with the LD dye LipiBlue under *luci* and *hxk-2i* (N=3). Scale bars, 10 μm. (**D**) Intensity plot of the region of interest (ROI) from *hxk-2i* (**C**). (**E**) Co-localization of MML-1::GFP and LD in *luci* and *hxk-2i*. The co-localization analysis was calculated with the thresholded Manders' co-localization coefficients tM1 (GFP channel) and tM2 (LipiBlue channel) (N=3). (**F**) Quantification of LD diameter in day 1 adult worms expressing the LD reporter PLIN-1::mCherry upon knockdown of *hxk-2* and *tkt-1* (N=3). (**G**) Quantification of MML-1 nuclear localization under knockdown of the LD metabolic enzymes *dgat-2* and *atg-1* (N=3). (**H**) Quantification of MML-1 nuclear localization upon double knockdown of *tkt-1* and *atgl-1* (N=3). Statistical significance was calculated with a one-way ANOVA test in E–H. (**E–H**) bars represent mean ± SD.

The online version of this article includes the following source data and figure supplement(s) for figure 4:

**Source data 1.** Raw data from *Figure 4*.

**Figure supplement 1.** MML-1 interactome is enriched with proteins from different organelles.

**Figure supplement 1—source data 1.** Raw data from *Figure 4—figure supplement 1*.

Therefore, we wondered whether the LD size correlates with MML-1 nuclear localization. We knocked down *hxk-2* and *tkt-1* to decrease and increase MML-1 nuclear localization, respectively. We observed a significant reduction in diameter and the total amount of LDs upon knockdown of *tkt-1*, while *hxk-2i* did not have an effect (*Figure 4F*, *Figure 4—figure supplement 1H*). Double knockdown of *hxk-2* and *tkt-1*, however, resulted in smaller and fewer LDs than the control. We also wanted to test whether

the HXK-1/fatty acid β-oxidation axis regulation of MML-1 also converged on LD size regulation. Interestingly, we found that knockdown of *acs-2i* and *hxk-1i acs-2i* double knockdown resulted in a mild but significant increase in LD size (*Figure 4—figure supplement 1I*), supporting the notion that the two hexokinases regulate MML-1 via distinct mechanisms. Collectively, these data suggest that MML-1 can localize with a specific subpopulation of mitochondria associated with LD and that inhibition of the PPP can rescue MML-1 nuclear localization upon *hxk-2i* correlated with decreasing LD size.

To test whether LDs directly regulated MML-1 localization, we knocked down *dgat-2* and *atgl-1*, two enzymes involved in LD metabolism. DGAT-2 is a diacylglycerol transferase that promotes LD droplet expansion, while ATGL-1 is a rate-limiting enzyme in LD breakdown (*Figure 4B*). First, we measured the neutral lipid composition and found that *atgl-1i* increased lipid levels while *dgat-2i* worms did not show significant changes under ad libitum conditions (*Figure 4—figure supplement 1J*). Next, we tested MML-1 nuclear localization and observed that *dgat-2i* increased MML-1 nuclear localization while *atgl-1i* resulted in decreased levels of nuclear MML-1 (*Figure 4G*). Finally, we did an epistatic analysis of LD metabolism and the PPP and found that MML-1 nuclear localization in *atgl-1i* was rescued upon *tkt-1* knockdown (*Figure 4H*), thus suggesting that LD metabolism is upstream or parallel to the PPP in MML-1 regulation.

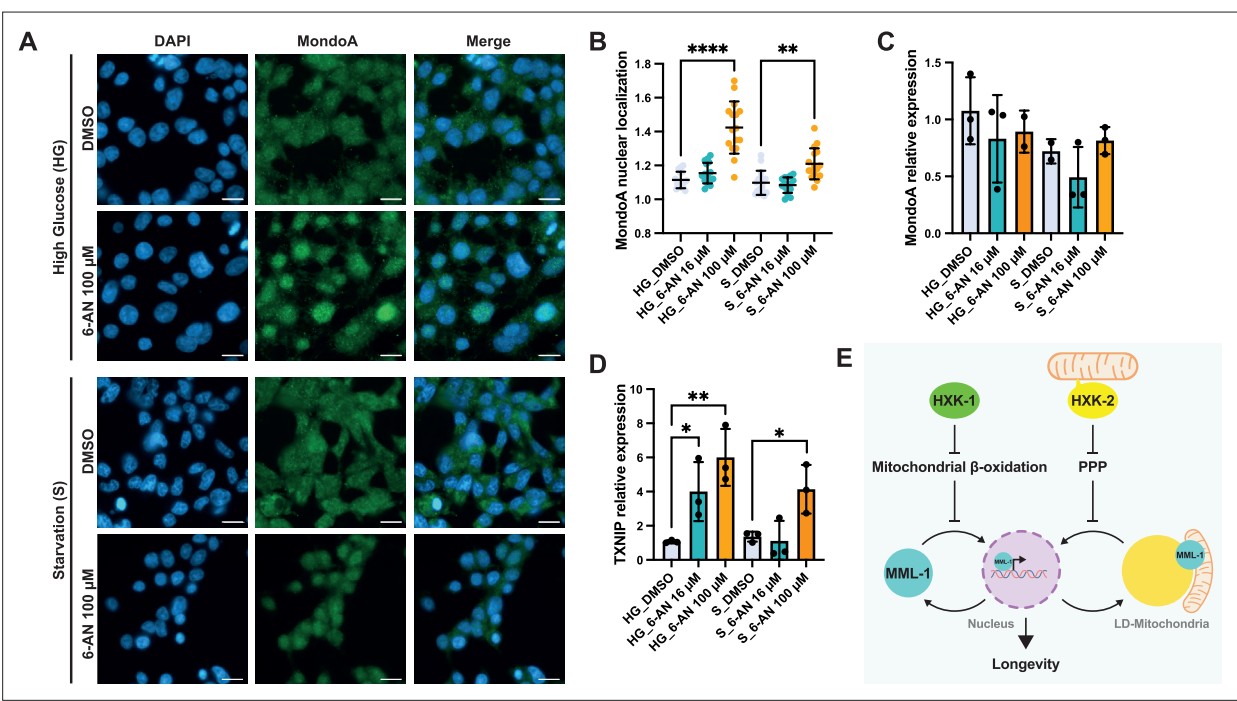

**Figure 5.** PPP regulates MondoA localization and transcriptional function in HEK293T cells. (**A**) Immunofluorescence of MondoA in HEK293T cells under high glucose media (HG) and starvation (S) conditions supplemented with the PPP inhibitor 6-aminonicotinamide (6-AN) 100 μM. Scale bars, 20 μm. (**B**) Quantification of MondoA nuclear localization in HEK293T cells treated with 6-AN 16 and 100 μM under HG and S conditions (N=3). (**C**) Relative mRNA levels of MondoA in HEK293T cells treated with 6-AN 16 and 100 μM under HG and S conditions measured by qPCR (N=3). (**D**) Relative mRNA levels of MondoA downstream target *TXNIP* in HEK293T cells treated with 6-AN 16 and 100 μM under HG and S conditions measured by qPCR (N=3). Statistical significance was calculated with one-way ANOVA in B–D. (**B**) Bars represent mean ± SD, (**C, D**) bars represent mean ± SEM. (**E**) Working model of MML-1 regulation by hexokinases. HXK-1 and HXK-2 positively regulate MML-1 through different downstream mechanisms. On the one hand, mitochondrial β-oxidation inhibition rescues MML-1 nuclear localization upon *hxk-1i*. On the other hand, the PPP regulates MML-1 subcellular localization between LD mitochondria and the nucleus under *hxk-2i*, which is required for the lifespan extension mediated by germline loss. PPP, pentose phosphate pathway; LD, lipid droplet.

The online version of this article includes the following source data and figure supplement(s) for figure 5:

**Source data 1.** Raw data from *Figure 5*.

**Figure supplement 1.** Inhibition of pentose phosphate pathway (PPP) NADPH-producing enzymes by 6-aminonicotinamide (6-AN) induces MondoA downstream targets.

**Figure supplement 1—source data 1.** Raw data from *Figure 5—figure supplement 1*.

## The PPP regulates MondoA function in mammalian cells

We next asked whether the PPP regulation of MML-1 we observed in *C. elegans* was conserved in mammals. Specifically, we wanted to determine whether inhibition of the PPP is sufficient to induce MondoA translocation to the nucleus under contrasting conditions where MondoA is mostly cytosolic (starvation media) or split between the nucleus and cytoplasm (high-glucose growing media). To test this, we cultured the HEK293T cells in high-glucose media with 6-AN and subsequently transferred the cells to either high-glucose media or starvation media (*Figure 5—figure supplement 1A*). First, we observed decreased NADPH/NADP$^+$ ratio in cells treated with 6-AN and under starvation conditions (*Figure 5—figure supplement 1B*). Next, we found that in both conditions, MondoA nuclear localization was increased upon inhibition of the PPP (*Figure 5A and B*), while there was no evident regulation of MondoA at the transcript level (*Figure 5C*). Consistent with MondoA increased nuclear localization, we observed increased expression of two of the best-described downstream targets of MondoA, *TXNIP* and *ARRDC4* (*Figure 5D*, *Figure 5—figure supplement 1C*). These findings indicate that the PPP regulates mammalian MondoA localization and transcriptional function.

## Discussion

Nutrient sensing pathways and metabolism play a central role in modulating aging by establishing resilience and survival metabolic states. Although much work has focused on identifying mediators of these processes, we still know little about how organisms coordinate feedback loops of metabolic pathways and transcriptional responses to establish such states. Here, we present novel carbohydrate and lipid metabolism mechanisms regulating the transcription factor MML-1/Mondo. We observed that MML-1 senses multiple steps of glucose metabolism and found that hexokinases regulate MML-1 function through different mechanisms in *C. elegans*. On the one hand, we found that under *hxk-2* knockdown, MML-1 re-localizes to mitochondria and LD, abolishing the longevity of germline-less worms and mutants with reduced mitochondrial function. Moreover, we could rescue MML-1 function by decreasing the PPP independent of the G6P levels. On the other hand, we found that *hxk-1* knockdown decreased MML-1 function by upregulating mitochondrial β-oxidation. Thus, we found two metabolic branches that converge on MML-1 to trigger a transcriptional response and regulate longevity. We note that while MML-1 is required in four different longevity models, namely *glp-1, isp-1, daf-2,* and *raga-1, hxk-1* and *hxk-2* were required only for *glp-1* and *isp-1* mediated longevity. Presumably, the signal transduction wiring is different in *raga-1* and *daf-2*, or their inputs into MML-1 work downstream or in parallel to the hexokinases.

MondoA and ChREBP transcription factors have well-established central roles in glucose metabolism (*Ma et al., 2006*; *Sans et al., 2006*; *Stoltzman et al., 2008*; *Yamashita et al., 2001*), but whether and how glucose metabolic circuits affect Mondo regulation in a whole animal model is little explored. By performing a systematic RNAi-based targeted screen of enzymes involved in glucose anabolism and catabolism, we found that MML-1 senses multiple steps of glucose metabolism in the worm. Among the genes tested, we found MML-1 nuclear localization to be broadly sensitive to the downregulation of glycolytic enzymes, and the knockdown of hexokinase had the most profound effect. Hexokinase carries out the first step in glucose metabolism, the phosphorylation of glucose to form G6P, which can be shunted to multiple metabolic pathways, including glycolysis, the PPP, glycogenesis, and trehalose production. Our findings are consistent with previous work in cultured cells showing that MondoA localization is regulated by HKII, which is thought to stimulate nuclear localization via the production of G6P (*Sans et al., 2006*). Additionally, MondoA senses other phosphorylated hexoses, including allose and 3-*O*-methylglucose (*Stoltzman et al., 2011*).

Consistent with our genetic findings above, pharmacological inhibition of glycolysis by supplementation with 3-BrP and 2-DG suppressed MML-1 nuclear localization. In contrast to MML-1, MondoA has been shown to accumulate in the nucleus upon 2-DG in rat L6 cells (*Stoltzman et al., 2008*). A possible explanation for this difference is that our treatment represents a chronic exposure to 2-DG from development until adulthood, while the experiments in cell culture may reflect an acute response. Overall, 2-DG has been used as a glycolytic inhibitor because it can be phosphorylated by hexokinase to 2-DG 6-phosphate; however, it cannot be further metabolized in glycolysis. Further evidence shows that 2-DG not only functions as a catabolic inhibitor but can also be directed to the PPP or other pathways (*Chi et al., 1987*; *Ralser et al., 2008*). Moreover, low concentrations of 2-DG

have been shown to extend lifespan by restricting glucose metabolism, while higher concentrations of 2-DG and glucose shortened the lifespan in worms (*Schulz et al., 2007*; *Lee et al., 2009*), suggesting a threshold for the benefits of glucose metabolism inhibition in lifespan.

Additionally, we found an important link between glycolysis and pyruvate metabolism regulating MML-1 localization. Under aerobic conditions, pyruvate is imported into the mitochondria and converted to acetyl-CoA to generate ATP and other reducing molecules. Downregulation of the mitochondrial pyruvate carrier *mpc-1* and the pyruvate dehydrogenase complex (*Supplementary file 1A*) also decreased MML-1 nuclear localization and function. Further, we found MPC-1 enriched in the MML-1 interactome. Interestingly, loss of *MPC1* in yeast has been shown to accumulate pyruvate, lower TCA cycle intermediates, and reduce chronological aging (*Bricker et al., 2012*; *Orlandi et al., 2014*). *MPC1* similarly regulates the disposition of glycolytic and TCA intermediates in flies and mammals (*Bricker et al., 2012*). Pyruvate can also enter the TCA cycle through pyruvate carboxylase PYC-1, which converts pyruvate to oxaloacetate. However, knockdown of *pyc-1* did not affect MML-1 localization (*Supplementary file 1A*), suggesting that MML-1 may sense pyruvate metabolism mainly through oxidative decarboxylation of acetyl-CoA.

We also found that MML-1 responds strongly to alterations in the TCA cycle. The TCA cycle allows organisms to oxidize carbohydrates, fatty acids, and amino acids to provide energy and intermediates for the biosynthesis of macromolecules. Interestingly, the knockdown of two subunits of the OGDC, *ogdh-1* and *dld-1*, increased MML-1 nuclear localization. The OGDC is a rate-limiting enzyme within the mitochondrial TCA cycle that decarboxylates 2-oxoglutarate to succinyl-CoA. It has been shown that supplementation of 2-oxoglutarate and knockdown of *ogdh-1* extends lifespan through inhibition of ATP synthase, decreasing oxygen consumption, and increasing autophagy in an mTOR-dependent manner (*Chin et al., 2014*). Whether MML-1 is required for this lifespan extension has yet to be explored. Conceivably, the mitochondrial localization of MML-1 could help sense 2-oxoglutarate levels, and the accumulation of this metabolite could signal translocation to the nucleus. Moreover, glutamine can be converted to glutamate and subsequently to 2-oxoglutarate during glutaminolysis and represents an important mechanism to replenish the TCA cycle under different metabolic challenges. In this case, OGDC plays a pivotal role in channeling anaplerotic reactions into the TCA cycle from glutamine and glutamate and other amino acids that can ultimately be converted to 2-oxoglutarate (i.e. histidine, proline, and arginine) (*Owen et al., 2002*). Interestingly, increased glutamine-dependent anaplerosis has been shown to regulate MondoA activity in BxPC-3 cells (*Kaadige et al., 2009*), though the molecular mechanism remains unclear. Whether glutamine, glutamate supplementation, or knockdown of glutamate dehydrogenase can regulate MML-1 nuclear localization remains to be determined.

On the other hand, the TCA cycle can operate in reverse to restore TCA intermediate levels (*Dalziel and Londesborough, 1968*). Increased levels of 2-oxoglutarate due to inhibition of OGDC or glutamine anaplerosis can undergo reductive carboxylation by isocitrate dehydrogenase to produce isocitrate. After that, aconitase can convert isocitrate to citrate, which can be exported from the mitochondria for de novo lipogenesis. MML-1 could help orchestrate the metabolic rewiring of the TCA cycle with the fatty acid metabolism transcriptional response. Notably, ChREBP has been shown to regulate many lipogenic enzymes, including acetyl-CoA carboxylase, the fatty acid synthase, and ATP-citrate lyase (*Ishii et al., 2004*; *Postic et al., 2007*). Accordingly, the MML-1 transcriptome shows clear regulation of several fatty acid metabolic enzymes (*Nakamura et al., 2016*).

We observed that PPP enzymes strongly influenced MML-1 nuclear localization and function. The PPP is a cytosolic pathway involved in the interconversion of sugars to generate precursors for the biosynthesis of lipids, amino acids, and nucleotides and maintain redox homeostasis. We found that knockdown of PPP components, PGDH *T25B9.9* and transketolase *tkt-1*, enhanced MML-1 nuclear localization, increased the transcription of MML-1 target genes, and restored longevity to *glp-1(e2141)* animals under *hxk-2i*. The PPP is the primary source of NADPH, which plays a vital role in many cellular processes, including fatty acid, nucleotide, neurotransmitter, and cholesterol metabolism, and is an essential reducing agent (*Ju et al., 2020*). We found that pharmacological inhibition with 6-AN of the two enzymes producing NADPH in the PPP increased MML-1 nuclear localization. We also found increased levels of NADPH and lower levels of G6P under *hxk-2* knockdown. Importantly, these observations suggest that residual G6P may be diverted to the PPP under low mitochondrial hexokinase and further imply that metabolites other than G6P could regulate MML-1 nuclear localization. What

this other metabolite could be remains elusive, but a simplifying notion is that MML-1 nuclear localization is promoted by glycolysis or PPP intermediates. Blocking entry into the PPP or TCA might be expected to accumulate such metabolites and promote Mondo nuclear localization. Conversely, depletion of these metabolites would disrupt Mondo nuclear localization and activity.

Another possibility is that Mondo organellar distribution is regulated at another scale. Notably, a major output of the PPP and NADPH metabolism is the production of lipids. Interestingly, we found that under *tkt-1* knockdown, fat levels decreased, and LDs were smaller in diameter, suggesting that a reduction in LDs could signal MML-1 to translocate to the nucleus. Consistently, transaldolase *tald-1* knockdown has been shown to increase lifespan and decrease intestinal fat levels in the worm through increased activity of adipose triglyceride lipase ATGL-1 (*Bennett et al., 2017*). This enzyme is involved in the metabolism of LD under starvation to mobilize stored fats for energy production (*Lee et al., 2014*).

Interestingly, although both HXK-1 and HXK-2 positively regulate MML-1 function, they do so through different mechanisms (*Figure 5E*). We observed that decreasing the PPP rescued MML-1 nuclear localization and *glp-1* longevity under *hxk-2* knockdown, independent of the G6P levels, but not under *hxk-1* knockdown. Conversely, inhibition of mitochondrial fatty acid β-oxidation is sufficient to rescue MML-1 nuclear localization under *hxk-1* knockdown, but not under *hxk-2* knockdown. Interestingly, *acs-2* knockdown abolished *glp-1* longevity (data not shown), consistent with previous work showing that NHR-49, a transcription factor that drives *acs-2* expression, is required for *glp-1* longevity (*Ratnappan et al., 2014*). Thus, inhibiting fatty acid β-oxidation promotes MML-1 nuclear localization under *hxk-1i* but abolishes lifespan extension, potentially due to epistatic effects on other transcription factors or processes.

How both hexokinases activate different metabolic pathways to regulate MML-1 remains unknown and could work at multiple levels. First, there is evidence that differences in the subcellular location of hexokinases may result in the compartmentalization of glucose metabolism, with the channeling of G6P to different pathways (*John et al., 2011*; *Wilson, 2003*). Mitochondrial hexokinase has preferential access to ATP generated in the mitochondria and provides efficient glycolysis coupling with further pyruvate oxidation by the TCA cycle and oxidative phosphorylation. Furthermore, mammalian HKI and HKII have a dynamic localization, shuttling between the mitochondrial outer membrane and the cytoplasm (*John et al., 2011*). Both mammalian hexokinases have been reported to reversibly bind to mitochondria through their N-terminal sequence (*Roberts et al., 2014*) and via their interaction with VDAC1 (*Lindén et al., 1982*). This dual localization allows the cell to adapt to different metabolic requirements and maintain energetic balance. Under high levels of G6P, HKII shuttles from the mitochondria to the cytoplasm, causing cells to use glycogen as an energy source (*John et al., 2011*). The subcellular localization also plays other significant roles in different signaling pathways. For example, mitochondria-bound hexokinase has anti-apoptotic effects (*Gottlob et al., 2001*), and increased *N*-acetylglucosamine upon pathogen exposure releases hexokinase from the mitochondria and activates innate immune responses (*Wolf et al., 2016*). Here, we found that *C. elegans* HXK-2 is associated with mitochondria and that this localization is abolished by deleting the first 10 amino acids. However, we saw no effect upon *vdac-1* knockdown nor dynamic localization of HXK-2 between mitochondria and cytosol (data not shown). Taken together, these observations suggest that the nematode has dedicated mitochondrial (HXK-2) and cytosolic (HXK-1 and HXK-3) hexokinases.

We also found differences in the tissue expression of the different hexokinase isozymes of the nematode: HXK-1 was mainly found in the pharynx, neurons, gonadal sheath, and coelomocytes, while HXK-2 was present in the hypodermis, body wall muscle, and intestine. Consistently, RNAseq data from different tissues in *C. elegans* shows that *hxk-2* is the main isozyme in the body wall muscle and intestine (*Hutter and Suh, 2016*). At the same time, *hxk-1* is the main one expressed in pharyngeal muscle and neurons. This differential distribution could explain how *hxk-1* and *hxk-2* differentially regulate MML-1 through β-oxidation and PPP pathways. Notably, we scored MML-1 nuclear localization in the intestine in our experiments, where *hxk-2* but not *hxk-1* is expressed. In the future, it will be important to see whether *hxk-1* cell autonomously impacts *mml-1* nuclear localization within the same tissue, such as the pharynx.

In any case, our observation that HXK-1 affects intestinal MML-1 nuclear localization suggests cell non-autonomous signaling. What might be the nature of such cell non-autonomous signaling? One possibility is the sterol hormone signaling pathway, which acts in somatic reproductive tissue

to regulate *glp-1(e2141)* longevity. Decreasing the activity of the nuclear hormone receptor DAF-12, or its ligands dafachronic acids, abolishes the longevity of germline-less worms (*Gerisch et al., 2001*; *Yamawaki et al., 2010*). Interestingly, carbohydrate metabolism directly impacts the production of dafachronic acids as NADPH is required for the last step of dafachronic acid biosynthesis by DAF-9 (*Motola et al., 2006*; *Penkov et al., 2015*). Furthermore, DAF-9 is expressed in the somatic gonad (spermatheca) and XXX neurons in adult animals (*Gerisch and Antebi, 2004*). Conceivably, hexokinase downregulation could affect the production of NADPH and dafachronic acids regulating DAF-12 activity, thereby affecting MML-1 function. It will be interesting to see whether dafachronic acid supplementation can rescue MML-1 nuclear localization upon *hxk-1i* or *hxk-2i*.

The nervous system also plays a crucial role in integrating metabolic status and systemic responses in a cell non-autonomous manner (*Mutlu et al., 2020*; *Zhang et al., 2019*). Inhibition of the mTOR signaling pathway has been shown to extend lifespan across the evolutionary spectrum (*Kennedy and Lamming, 2016*), and recent work has shown that restoring *raga-1*/RagA or *rsks-1*/S6K in the nervous system is enough to entirely suppress the lifespan extension conferred by mutation of these mTOR signaling factors (*Zhang et al., 2019*). In cells, HKII has been shown to shuttle from the mitochondria and directly interact with mTOR under glucose deprivation and inhibit its function, resulting in an induction of autophagy (*Roberts et al., 2014*). This interaction is dependent on the mTOR signaling (TOS) motif present in HKII required for binding to Raptor. Interestingly, this TOS motif (KDIDI) is conserved in the nematode HXK-1; however, HXK-2 and HXK-3 lack this sequence. Our finding that *raga-1* mutants are epistatic to hexokinase knockdown for lifespan extension is consistent with a role of mTOR signaling downstream of hexokinase. It would be interesting to study whether the interaction between hexokinase and mTOR is conserved in *C. elegans* and whether decreasing the main neuronal hexokinase (HXK-1) would result in mTOR-dependent inhibition of MML-1.

Proteins tend to co-localize near their substrates to maximize specificity and regulate signaling pathways. In particular, MondoA was first reported to be associated with the mitochondria outer membrane in primary human SkMC cells through protein-protein interaction (*Sans et al., 2006*). We have previously shown that this localization is conserved in *C. elegans* (*Nakamura et al., 2016*). Mitochondria-bound MondoA has been proposed to have preferential access to newly synthesized G6P from HKII to coordinate metabolism and transcriptional response (*Wilde et al., 2019*). However, the identity of the mitochondrial factor tethering MondoA at the mitochondria remains unknown.

Interestingly, our proteomics analysis of MML-1 IP showed enrichment with proteins from different cellular compartments, including mitochondria, ER, LD, and nucleolus, suggesting an important role of Mondo in integrating diverse organellar signals. Localization to mitochondria was particularly conspicuous. Multiple nuclear transcription factors are localized in the mitochondria, although dissecting their nuclear and mitochondrial function is challenging (*Leigh-Brown et al., 2010*). On the one hand, nuclear transcription factors have been shown to directly bind the mitochondrial genome to regulate transcription, such as the cAMP response element-binding protein (*Marinov et al., 2014*). Interestingly, *mml-1(ok849)* loss-of-function mutants showed a decrease in all 12 mitochondrial-encoded genes compared to wild type (*Nakamura et al., 2016*). However, this could be indirect by regulating the mitochondrial transcription factor TFAM or mitochondrial biogenesis (*Li et al., 2005*). On the other hand, transcription factors like p53 have been shown to translocate to the mitochondrial outer membrane under apoptotic signals to interact with Bcl-2 and promote membrane permeabilization and apoptosis (*Marchenko et al., 2007*). Future studies will help elucidate whether MML-1/MondoA mitochondrial localization regulates other cellular processes independent of its nuclear transcription function.

LD represents the main lipid storage in cells to balance metabolism and energy demands. These organelles are essential for many signaling pathways and are required for survival during fasting by mobilizing lipids for energy production. In *C. elegans*, LDs are mainly localized in the intestine and hypodermis and are composed almost exclusively of TAG (*Vrablik et al., 2015*). We found PLIN-1, one of the most abundant LD proteins, enriched in the MML-1 interactome. Interestingly, we found that MML-1 co-localizes with LD in vivo in the intestine. MondoA was previously seen in large-scale proteomics associated with LD (*Krahmer et al., 2018*). It was shown that MondoA and Mlx localized with LD through amphipathic helices in the C-terminus in SUM159 cells (*Mejhert et al., 2020*). The researchers proposed a model in which MondoA/Mlx co-localization with LD inhibits its transcriptional activity, limiting glucose-dependent transcription. Work by Brunet and colleagues suggests that

MUFA intercalation into LD and peroxisomes somehow promotes longevity, though downstream transcription factors are little explored (*Papsdorf et al., 2023*). Mitochondria and LDs physically interact with one another to regulate metabolism (*Boutant et al., 2017*). As MondoA has also been shown to localize with mitochondria, it would be interesting to understand the role of this organelle in regulating MondoA's association with LD. Although the proteins involved in these contact sites are just beginning to be uncovered, both PLIN1/PLIN-1 and ACSL1/ACS-13 have been proposed to tether mitochondria and LD (*Gordaliza-Alaguero et al., 2019*).

Mitochondria bound to LDs facilitate the coordination of TAG metabolism and fatty acid oxidation. For example, lipases (e.g. ATGL) in the LDs break down TAG to release free fatty acids that can be activated in the mitochondria's outer membrane by ACSL1 to fatty acyl-CoA. This acyl-CoA can be imported into the mitochondria for β-oxidation. We propose a model (*Figure 5E*) in which MML-1 interacts with specific mitochondria associated with LD to orchestrate a transcriptional response depending on fuel utilization. Under normal conditions, glucose stimulates MML-1 translocation to the nucleus to activate the transcription of glycolytic enzymes. In contrast, upon blockage of glycolysis (e.g. hexokinase knockdown), the cell utilizes fatty acids to produce ATP, re-localizing and inhibiting MML-1 at the mitochondria associated with LD.

## Materials and methods

### Worm growth and RNAi treatment

*C. elegans* worm strains were grown and maintained on nematode growth medium (NGM) plates seeded with *Escherichia coli* OP50 strain as the primary food source. Hermaphrodite worms were used in all experiments. Worms were grown at 20°C unless stated otherwise. *glp-1(e2141)* strains were maintained at 15°C. All worm strains are listed in *Supplementary file 1D*. Worms were synchronized by 4–6 hr egg lays at 20°C. For experiments including *glp-1(e2141)*, egg lays were performed at 15°C, and the eggs were upshifted to 25°C for 52–56 hr to induce the germline-less phenotype. After *glp-1* induction, worms were downshifted to 20°C until the day of analysis or collection. All materials and worm strains generated are available upon request.

The NGM for RNAi knockdown was supplemented with 1 mM isopropyl β-1-D-thiogalactopyranoside and 100 µg/mL ampicillin. The plates were seeded with the RNAi-expressing *E. coli* HT115 (DE3) strain transformed with the L4440 vector from the Ahringer and Vidal libraries (*Kamath and Ahringer, 2003*; *Rual et al., 2004*) or cloned for this work. Bacteria containing luciferase RNAi cloned into the L4440 vector were used as controls. Worms were grown on RNAi plates egg-on in all the experiments.

### Cell culture experiments

HEK293T (ATCC, CRL-3216) cells were maintained in DMEM 4.5 g/L D-glucose (Gibco) supplemented with 10% FBS (Gibco). Cell identity was determined by STR profiling, and Mycoplasma contamination was determined by sequencing (Eurofins Genomics). For blocking the PPP, cells were supplemented with 16 and 100 µM 6-AN for 24–48 hr. After, cells were washed and added fresh DMEM or transferred to starvation media (NaCl 140 mM, CaCl$_2$ 1 mM, MgCl$_2$ 1 mM, HEPES pH 7.4 20 mM, bovine serum albumin [BSA] 1%) for 6 hr. Cells were collected for immunofluorescence, RNA extraction, and metabolite quantification.

For addressing MondoA localization, cells were fixed with 4% paraformaldehyde (PFA) for 15 min at room temperature (RT). Following fixation, the samples were washed three times with 500 µL DPBS, followed by 500 µL of methanol 99% for 5 min at –20°C, and washed twice with 500 µL of DPBS and three times with TBST 1X. Cells were blocked with 10% goat serum diluted in DPBS for 1 hr at RT. Cells were incubated with 200 µL of the Anti-MondoA antibody (Bethyl Laboratories, A303-195A) 1:500 diluted in goat serum overnight at 4°C. Next, cells were washed three times with DPBS containing Triton X-100 (PBST) and similarly incubated with 200 µL of Alexa-488 secondary antibody (Invitrogen, A21206) 1:500 diluted in the goat serum at RT and covered in aluminum foil for 2 hr. After washing three times with 500 µL of PBST, the samples were covered with a mounting medium containing DAPI. The cells were cured for 24 hr at RT in the dark and then stored at 4°C.

## Demographic analysis

Lifespan analyses were conducted by synchronizing worms by egg lay. All animals were kept at 20°C egg-on, or transferred to 20°C on day 1 of adulthood for the experiments, including *glp-1(e2141)*. More than 120 worms were used per strain/condition with 20–30 worms per 6 cm plate. Worms were transferred every other day to fresh plates until they stopped laying eggs and were scored every time they were transferred. Animals with internal hatching and vulva protrusions were censored from the analysis. All lifespans were performed by blinding the genotypes/RNAi. Demographic parameters like mean, median, and maximum lifespan were calculated and plotted using GraphPad Prism 9 software.

## OCR measurements

OCR measurements in *C. elegans* were based on *Koopman et al., 2016*. Worms were synchronized by egg lay. 10–15 adult worms were transferred to a Seahorse XF96 Cell Culture Microplate (Agilent) with 200 µL of M9 buffer. OCR was measured using the Seahorse XFe96 Analyzer (Agilent). To measure mitochondrial fatty acid β-oxidation, basal oxygen consumption was measured followed by injection of etomoxir (final well concentration 500 µM) and finally injection of sodium azide (final well concentration 40 mM), all measured five times. OCR was normalized to the number of worms with at least six technical replicates.

## Body measurements, pharyngeal pumping rates, and motility assay

Worms were imaged or recorded using the Leica stereo microscope MDG41 equipped with Leica DFC3000G monochrome camera with a ×6.3 magnification. For measuring the body length and area, >20 worms were transferred to a drop of sodium azide 50 mM, imaged, and analyzed in Fiji imaging software. To measure the pharyngeal pumping rates, worms were transferred to a new plate, acclimated on the plates without a lid for 10 min, and then recorded for 30 s. Worm motility was assessed by placing >20 worms in a drop of M9 buffer and recording for 30 s using the Leica M80 Binocular Microscope equipped with the Leica MC170 HD camera. Videos were analyzed to count the thrashing of each worm.

## Molecular cloning

Plasmids were constructed by classical cloning or Gibson Assembly. Restriction enzymes from NEB were used according to the manufacturer's instructions. Custom primers were designed with Snap-Gene v.5.3.2 and NEBuilder Assembly Tool v.2.5.4 (*Supplementary file 1E*). For classical cloning, digested plasmids and amplicons were ligated using the T4 DNA Ligase according to the manufacturer's instructions (New England Biolabs). For Gibson Assembly, digested plasmid and amplicon were assembled using NEBuilder HiFi DNA Assembly Master Mix (New England Biolabs) according to the manufacturer's instructions. Plasmids were transformed into *E. coli* DH5α (Life Technologies) strain using the heat shock protocol and plated on LB media supplemented with ampicillin.

## RNA extraction and RT-qPCR

Worms were synchronized by egg lay for 4 hr and collected once they reached the young adult stage in 15 mL conical tubes. The worm pellet was subjected to four cycles of freeze and thaw with liquid nitrogen and a water bath at ~37°C. To lyse the worms, 200 µL of 1.0 mm Zirconia/Silica beads (Fisher Scientific) were added and transferred to the TissueLyser LT (QIAGEN) for 30 min, full speed at 4°C. Samples were incubated for 5 min at RT. After, 200 µL chloroform was added and centrifuged at 12,000×$g$ for 15 min at 4°C. The top aqueous phase was transferred to a new 1.5 mL tube and mixed with 200 µL of ethanol. The RNA purification was performed using the RNeasy Mini Kit (QIAGEN) according to the manufacturer's instructions. The concentration and purity of the RNA samples were quantified using the NanoPhotometer NP80 spectrophotometer (Implen).

cDNA was prepared after extracting total RNA from young adult worms or cell extracts using the iScript cDNA Synthesis Kit (Bio-Rad) according to the manufacturer's instructions. Primers, cDNA, and Power SYBR Green Master Mix (Applied Biosystems) were transferred to a 384-well plate using the JANUS automated workstation (PerkinElmer) with four technical replicates per genotype/gene. The reaction was quantified using the ViiA 7 Real-Time PCR system machine (Applied Biosystems). *cdc-42* was used as an endogenous control for the quantification. The complete primer sequences are given in *Supplementary file 1E*.

## Oxidative stress assay

Worms were grown on NGM plates until day 1 adults and then transferred to NGM supplemented with paraquat at 15 and 35 mM for 6 hr. As a control, we used the CL2166 strain that contains the transcriptional reporter *gst-4p*::GFP. MML-1::GFP nuclear localization was measured in the intestine.

## Protein extraction and WB analysis

Worms were harvested from a mixed population or synchronized by egg lay or bleaching with M9 buffer and washed three times. The worm pellet was snap-frozen in liquid nitrogen and stored at –80°C until extraction. The pellet was resuspended in liquid lysis buffer supplemented with cOmplete ULTRA EDTA-free protease inhibitors (Roche) and PhosSTOP phosphatase inhibitors (Roche) and sonicated with Bioruptor Plus (Diagenode S.A.) coupled to Minichiller 300 (Huber) for 15 cycles of 30 s sonication, 30 s rest at 4°C. The total protein extract was cleared by spinning down at 20,817×g for 10 min at 4°C. Supernatant was transferred to a new tube, and protein was quantified using Pierce BCA Protein Assay Kit (Thermo Fisher Scientific) according to the manufacturer's instructions. Equal amounts of protein were mixed with 4× loading sample buffer containing DTT 50 mM and boiled at 95°C for 10 min. Samples were loaded onto an SDS-PAGE PERCENTAGE gel (Bio-Rad), and proteins were then transferred onto a nitrocellulose membrane (Bio-Rad) using the Trans-Blot Turbo Transfer System (Bio-Rad). Membranes were blocked for 1 hr at RT with 5% milk in TBST. Primary and secondary antibodies were diluted in 5% BSA in TBST and incubated for 2–18 hr at 4°C. Antibodies: RFP Tag Polyclonal Antibody 1:1000 (Thermo Fisher Scientific, R10367), Monoclonal ANTI-FLAG M2, Clone M2 1:1000 (Sigma-Aldrich, F1804), Anti-beta Actin Antibody 1:1000 (Abcam, Ab8224), Anti-rabbit IgG (H+L) 1:5000 (Invitrogen, G21234), Anti-mouse IgG 1:5000 (Invitrogen, G21040), Rabbit Anti-Mouse IgG (Light Chain Specific) (D3V2A) mAB (HRP Conjugate) 1:5000 (CST, 58802). The membranes were washed three times with TBST for 10 min at RT in between incubation with the antibodies and after the secondary. The signals were detected using Western Lightning Plus-ECL, Enhanced Chemiluminescence Substrate (PerkinElmer), and imaged in the ChemiDoc MP Imaging System with Image Lab software (Bio-Rad). Raw images are given in *Figure 1—figure supplement 1—source data 1* and *Figure 1—figure supplement 1—source data 2*.

## Immunoprecipitation

Worms were collected from a mixed population of twenty 10 cm plates, transferred into 15 mL conical tubes with M9 buffer, and washed three times. Worms were snap-frozen in liquid nitrogen and stored at –80°C until extraction. The worm pellet was resuspended in lysis buffer supplemented with cOmplete ULTRA EDTA-free protease inhibitors (Roche) and PhosSTOP phosphatase inhibitors (Roche). Worms were sonicated with Bioruptor Plus (Diagenode S.A.) coupled to Minichiller 300 (Huber) for 30 cycles of 30 s sonication, 30 s rest at 4°C. The total extract was cleared by spinning down at 20,817×g for 15 min at 4°C. Supernatant was transferred to a new tube, and protein was quantified using Pierce BCA Protein Assay Kit (Thermo Fisher Scientific) according to the manufacturer's instructions.

The Anti-FLAG M2 magnetic beads (Sigma) were resuspended and equilibrated with TBST buffer. A total of 7 mg of total protein extract in 800 µL was mixed with the magnetic beads and incubated overnight at 4°C. The next day, the supernatant was removed and stored for WB analysis. The beads were washed three times with 1 mL of cold washing buffer #1 (Tris/HCl, pH 7.4 20 mM, NaCl 300 mM, EDTA 1 mM, NP-50 0.5%), and three times with 1 mL of cold washing buffer #2 (Tris/HCl, pH 7.4 20 mM, NaCl 300 mM, EDTA 1 mM). In the last wash, 500 µL of washing buffer #2 was added, and 100 µL were stored for WB analysis.

To elute samples for mass spectrometry, 100 µL of elution buffer (Trypsin 5 ng/µL, Tris/HCl, pH 7.5 50 mM, Tris(2-carboxyethyl)phosphine 1 mM, chloroacetamide 5 mM) was added and incubated for 30 min at RT. The supernatant was transferred to a new 0.5 mL tube and incubated overnight at 37°C. The next day, formic acid (FA) was added to a final concentration of 1% to stop the digestion. Peptides were stored at –20°C until purification.

The peptides were cleaned and purified with StageTip. A 30 µg C18-SD tip was hydrated with 200 µL of methanol, followed by 200 µL of 40% acetonitrile (ACN)/0.1% FA. 200 µL of 0.1% FA was added to equilibrate the C18-SD. The digested peptides were dissolved in 0.1% FA and added to the C18-SD and washed once with 0.1% FA. To elute the peptides, 80 µL of 40% ACN/0.1% FA was

added, and elutes were collected in 0.5 µL tubes and dried in a centrifugal vacuum Concentrator Plus (Eppendorf) for 45 min at 45°C.

## LC-MS/MS analysis

Peptides were separated on a 25 cm, 75 µm internal diameter PicoFrit analytical column (New Objective) packed with 1.9 µm ReproSil-Pur 120 C18-AQ media (Dr. Maisch) using an EASY-nLC 1200 (Thermo Fisher Scientific). The column was maintained at 50°C. Buffer A and B were 0.1% FA in water and 0.1% FA in 80% CAN, respectively. Peptides were separated on a segmented gradient from 6% to 31% buffer B for 57 min and from 31% to 44% buffer B for 5 min at 250 nL/min. Eluting peptides were analyzed on an Orbitrap Fusion Tribrid mass spectrometer (Thermo Fisher Scientific). Peptide precursor m/z measurements were carried out at 60,000 resolution in the 350–1500 m/z range. The most intense precursors with charge states from 2 to 7 were selected for HCD fragmentation using 27% normalized collision energy. The cycle time was set to 1 s. The m/z values of the peptide fragments were measured at a resolution of 50,000 using an AGC target of 2e5 and 86 ms maximum injection time. Upon fragmentation, precursors were put on a dynamic exclusion list for 45 s.

The raw data were analyzed with MaxQuant v.1.6.1.0 (*Cox and Mann, 2008*). Peptide fragmentation spectra were searched against the canonical and isoform sequences of the *C. elegans* reference proteome (proteome ID UP000001940, downloaded December 2018 from UniProt). Methionine oxidation and protein N-terminal acetylation were set as variable modifications; cysteine carbamidomethylation was set as fixed modification. The digestion parameters were set to 'specific' and 'Trypsin/P'. The minimum number of peptides and razor peptides for protein identification was 1; the minimum number of unique peptides was 0. Protein identification was performed at a peptide spectrum match and protein false discovery rate of 0.01. The 'second peptide' option was on. Successful identification was transferred between the different raw files using the 'Match between runs' option. Label-free quantification (LFQ) (*Cox et al., 2014*) was performed using an LFQ minimum ratio count of 2. LFQ intensities were filtered for at least two valid values in at least one group and imputed from a normal distribution with a width of 0.3 and a downshift of 1.8. Differential expression analysis was performed using the *limma* package v.3.34.9 (*Ritchie et al., 2015*) in R v.3.4.3 (*R Core Team, 2017*).

## G6P and NADP$^+$/NADPH measurements and TKT activity assay

G6P and NADP$^+$/NADPH were measured from total worm extracts or cell extracts using the Glucose-6-Phosphate Assay Kit (Sigma MAK014) and the NADP/NADPH-Glo Assay (Promega G9081), respectively, following the manufacturer's instructions. The Transketolase Activity Assay Kit (Fluorometric) (Sigma MAK420) was used to measure TKT activity in worm extracts following the manufacturer's instructions. Samples were normalized to total protein concentration.

*glp-1(e2141)* strains were used to avoid signals from the germline. Briefly, worms were synchronized as mentioned previously, collected in a 15 mL conical tube, and washed three times with M9 buffer. Worms were snap-frozen in liquid nitrogen and sonicated using the Bioruptor Plus (Diagenode S.A.) coupled to Minichiller 300 (Huber) for 15 cycles of 30 s sonication, 30 s rest at 4°C. Worms' extracts were cleared by centrifugation at 20,817×*g* for 15 min at 4°C. Samples were always kept on ice. As mentioned, part of the samples was used to quantify protein, and the rest was used to quantify the metabolites.

For G6P colorimetric detection, samples were run in technical duplicates in a 96-well plate. The standard curve was made using the G6P Standard solution provided by diluting with ddH$_2$O to generate 0 (blank), 2, 4, 6, 8, and 10 nmol/well standards. Samples were combined with the reaction mix in each well and mixed for 30 min at RT, protecting the plate from the light. The absorbance was measured at 450 nm using the POLARstar OMEGA Plate Reader with Luminescence (BGM Labtech). For the analysis, all samples were corrected for the background. The standard curve was generated with the appropriate values obtained by the standards mentioned previously, and the amount of G6P in each sample was calculated with the standard curve. Values were normalized to protein levels.

For the NADP$^+$/NADPH bioluminescent assay, samples were run in technical duplicates in a 96-well plate. 50 µL of worm lysis in M9 buffer was incubated with 50 µL base solution with 1% DTAB. Samples were split into two: one to measure NADP$^+$ and the other to measure NADPH. Samples were incubated with or without 25 µL 0.4 N of HCl to measure NADP$^+$ or NADPH, respectively, at 60°C for 15 min, and then the plate was equilibrated for 10 min at RT. Next, 25 µL of Trizma base was added

to each well of acid-treated samples to neutralize the acid, and 50 µL of HCl/Trizma solution was added to each well of base-treated samples. An equal amount of the NADP/NADPH-Glo Detection Reagent was added to each well and mixed at RT for 30 min by protecting the plate from the light. The luminescence was recorded using the POLARstar OMEGA Plate Reader with Luminescence (BGM Labtech). Samples were corrected for background luminescence. The relative luminescent units were standardized to the protein concentration.

## Bright-field and fluorescent imaging

Bright-field and fluorescent images with higher resolution were acquired using the Zeiss Axio Imager Z1 Fluorescent Microscope equipped with the Digital Microscope Camera Axio Cam Mono 506 and the IcC5 true color camera. For fluorescent microscopy, the microscope used a Colibri 7 LED Light Source and the filter sets GFP (Set 38 HE), TR (Set 45), and TL. The software used was Zeiss Zen v2.3.69.1017. Bright-field and fluorescent images were also acquired with the Leica stereo microscope MDG41 equipped with Leica DFC3000G monochrome camera. The microscope was equipped with the Leica EL6000 external light source. The software used was the Leica Application Suite X v3.7.5.24914.

All dyes were added to the bacterial lawn, incubated egg-on, and protected from the light until the day of imaging. Final solutions were made by diluting in M9 buffer. LipiBlue (Dojindo) was resuspended in 100 µL of DMSO to make a 100 µM stock. The final concentration used was 1 µM. C1-BODIPY-C12 500/510 (Thermo Fisher Scientific) was dissolved in DMSO to make a 5 mM stock. The final concentration used was 1 µM. Worms were mounted on a slide with a 5% agarose pad using sodium azide 50 mM. Levamisole 2 mM was used as an anesthetic to image mitochondria.

Nuclear localization of MML-1::GFP was quantified in intestinal cells from day 1 adult worms. Worms were anesthetized with sodium azide 40 mM. At least 20 worms were imaged per genotype/condition. Quantification of the transcription factor nuclear localization was performed by manually selecting each nucleus and calculating the pixel intensity per area. The same area was used to calculate the cytosolic and background signal. Values were corrected by background subtraction. Nuclear localization was calculated by dividing the nuclear signal from the cytosolic signal.

## Confocal imaging

Confocal images were acquired with Leica TCS SP8 confocal microscope equipped with a white light laser and a ×63 1.4 NA oil objective. The microscope was also equipped with HyD detectors for fluorescent images and a PMT detector for the bright field. The LAS X Life Science software was used to acquire images and 3D reconstruction. When multiple fluorescent proteins or dyes were imaged, sequential scanning was used to reduce the bleed-through of the signal from one channel to another. All images used for co-localization analysis were deconvoluted with the installed LIGHTNING package to optimize the images and reduce background noise.

## Statistical analysis and software

The results are presented as mean ± SD or SEM, as indicated in the figure legends. The number of biological replicates is shown in the figure legends as 'N', while the number of worms is presented as 'n'. Before comparing groups, the data were tested for Gaussian distribution using the Kolmogorov-Smirnov normality test with Dallal-Wilkinson-Lilliefor p-value. For normally distributed data, unpaired t-test with Welch's correction or one-way ANOVA with Brown-Forsythe and Welch's corrections was calculated in GraphPad Prism 9 (GraphPad software). For lifespan analysis, p-values were calculated using the Log-rank (Mantel-Cox) test. Statistical data from all the lifespans are included in *Supplementary file 1B*. Statistical significance for qPCR experiments was calculated with the two-tailed t-test. For co-localization analysis, Costes' randomization was performed to determine statistical significance. Significance levels are ns = not significant $p > 0.05$, *$p < 0.05$, **$p < 0.01$, ***$p < 0.001$, ****$p < 0.0001$.

Image analysis was performed using the open-source Fiji package software. An additional JACoP (*Bolte and Cordelières, 2006*) plugin was used for co-localization analysis. Densitometry analyses were performed using GelAnalyzer v.19.1. Graphs and images were generated in GraphPad Prism 9, Adobe Illustrator, and BioRender.

## Acknowledgements

We would like to thank the CGC (University of Minnesota) and SunyBiotech for some of the strains used in this study. We also thank Katrin Wollenweber and Nadine Hochhard for their technical assistance and the Proteomics, Bioinformatics, and FACS & Imaging Core Facilities at the MPI AGE for their support. Special thanks to Özlem Karalay, Andrea Annibal, Roberto Ripa, Sarah Kreuz, and Victoria Martínez Miguel for the discussions and comments. This work was funded by the Max Planck Society and the Cologne Graduate School of Ageing Research (CGA).

## Additional information

### Funding

| Funder | Grant reference number | Author |
|---|---|---|
| Max Planck Society | | Shamsh Tabrez Syed<br>Maximilian Vonolfen<br>Anna Loehrke<br>Tim Droth<br>Ilian Atanassov<br>Adam Antebi |
| Cologne Graduate School of Ageing Research | | Raymond Laboy<br>Eugen Ballhysa<br>Klara Schilling |
| EU Erasmus | Fellowship | Marjana Ndoci |

The funders had no role in study design, data collection and interpretation, or the decision to submit the work for publication. Open access funding provided by Max Planck Society.

### Author contributions

Raymond Laboy, Conceptualization, Data curation, Formal analysis, Investigation, Methodology, Writing – original draft, Writing – review and editing; Marjana Ndoci, Shamsh Tabrez Syed, Maximilian Vonolfen, Eugen Ballhysa, Tim Droth, Klara Schilling, Anna Loehrke, Investigation; Ilian Atanassov, Data curation, Formal analysis, Investigation; Adam Antebi, Conceptualization, Resources, Supervision, Funding acquisition, Writing – original draft, Project administration, Writing – review and editing

### Author ORCIDs

Raymond Laboy ![ORCID] https://orcid.org/0000-0002-0375-1550
Ilian Atanassov ![ORCID] https://orcid.org/0000-0001-8259-2545
Adam Antebi ![ORCID] https://orcid.org/0000-0002-7241-3029

Reviewer #1 (Public Review): https://doi.org/10.7554/eLife.89225.4.sa1
Reviewer #2 (Public Review): https://doi.org/10.7554/eLife.89225.4.sa2
Author response https://doi.org/10.7554/eLife.89225.4.sa3

## Additional files

### Supplementary files
MDAR checklist

Supplementary file 1. Supplementary tables.

## Data availability

All data generated is included in the manuscript, including file sources of raw data.

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
