## [Editor Report · eLife Assessment]

This **important** study utilizes the nematode *C. elegans* and mammalian cell culture to investigate the role of MML-1/Mondo in conserved regulation of metabolism and aging. The evidence supporting the conclusions is **convincing** and covers a range of areas including localization, upstream pathways, and conservation. The paper will be of interest to a broad range of biologists studying aging, metabolism, and transcriptional regulation.

---

## [Referee Report · Reviewer #1 (Public Review)]

In this manuscript, Laboy and colleagues investigated upstream regulators of MML-1/Mondo, a key transcription factor that regulates aging and metabolism, using the nematode *C. elegans* and cultured mammalian cells. By performing a targeted RNAi screen for genes encoding enzymes in glucose metabolism, the authors found that two hexokinases, HXK-1 and HXK-2, regulate nuclear localization of MML-1 in *C. elegans*. The authors showed that knockdown of hxk-1 and hxk-2 suppressed longevity caused by germline-deficient glp-1 mutations. The authors demonstrated that genetic or pharmacological inhibition of hexokinases decreased nuclear localization of MML-1, via promoting mitochondrial β-oxidation of fatty acids. They found that genetic inhibition of hxk-2 changed the localization of MML-1 from the nucleus to mitochondria and lipid droplets by activating pentose phosphate pathway (PPP). The authors further showed that the inhibition of PPP increased the nuclear localization of mammalian MondoA in cultured human cells under starvation conditions, suggesting the underlying mechanism is evolutionarily conserved. This paper provides compelling evidence for the mechanisms by which novel upstream metabolic pathways regulate MML-1/Mondo, a key transcription factor for longevity and glucose homeostasis, through altering organelle communications, using two different experimental systems, *C. elegans* and mammalian cells. This paper will be of interest to a broad range of biologists who work on aging, metabolism, and transcriptional regulation.

---

## [Referee Report · Reviewer #2 (Public Review)]

Raymond Laboy et.al explored how transcriptional Mondo/Max-like complex (MML-1/MXL-2) is regulated by glucose metabolic signals using germ-line removal longevity model. They believed that MML-1/MXL-2 integrated multiple longevity pathways through nutrient sensing and therefore screened the glucose metabolic enzymes that regulated MML-1 nuclear localization. Hexokinase 1 and 2 were identified as the most vigorous regulators, which function through mitochondrial beta-oxidation and the pentose phosphate pathway (PPP), respectively. MML-1 localized to mitochondria associated with lipid droplets (LD), and MML-1 nuclear localization was correlated with LD size and metabolism. Their findings are interesting and may help us to further explore the mechanisms in multiple longevity models. The data support their proposed working model.

Comments on Revised Version (from the Reviewing Editor):

The authors have addressed the remaining concerns from both reviewers, adding textual information for reviewer 1 and testing the roles of hxk-1 and lipid oxidation in regulating lipid droplets for reviewer 2. Specifically, they find that knockdown of acs-2 and hxk-1 acs-2 double knockdown each result in a mild but significant increase in LD size. This result supports that the two hexokinases regulate MML-1 via distinct mechanisms, and is reflected in the updated model.

---

## [Author Response]

The following is the authors’ response to the previous reviews.

**Recommendations for the authors:**

**Reviewer #1:**
The authors addressed my previous concerns successfully. However, some critiques are addressed only in the response letter but not in the text (major comment 3, minor point 2). It will be great if they mention these in some parts of their manuscript.

Major 3: We now mention the effect of *acs-2i* on life span in the discussion, lines 475-480:

“Interestingly, acs-2 knockdown abolished glp-1 longevity (data not shown), consistent with previous work showing that NHR-49, a transcription factor that drives acs-2 expression, is required for glp-1 longevity (Ratnappan et al., 2014). Thus, inhibiting fatty acid β-oxidation promotes MML-1 nuclear localization under hxk-1i but abolishes lifespan extension, potentially due to epistatic effects on other transcription factors or processes.”

Minor 2: We now speculate on the differences concerning hxk-3 knockdown on MML-1 nuclear localization resulting from the low expression of hxk-3 in adults, lines 99-102:

“Among the three *C. elegans* hexokinase genes, hxk-1 and hxk-2 more strongly affected MML 1 nuclear localization in two independent MML-1::GFP reporter strains (Figure 1B, Supplementary Figure 1A), while hxk-3 had just a small effect on MML-1 nuclear localization, probably due to its low expression in adult worms (Hutter & Suh, 2016).”

**Reviewer #2:**
The authors have adequately addressed my previous concerns in their revised manuscript. However, I have one remaining minor concern regarding the link between lipid metabolism and MML-1 regulation. As proposed by the authors, HXKs modulate MML-1 localization between LD/mito and the nucleus. They have provided evidence supporting the roles of hxk-2 and the PPP in this regulatory process. Nonetheless, the involvement of hxk-1 and fatty acid oxidation (FAO) within this proposed framework remains unclear. Although FAO is generally believed to affect LD size, the potential effects of hxk-1 and FAO on LD should be investigated within the current study to further substantiate their model.

We thank the reviewer for this comment. We now examine how *hxk-1* and *acs-2* affect lipid droplet size. Interestingly, we found that knockdown of *acs-2* and *hxk-1 acs-2* double knockdown resulted in a mild but significant increase in LD size (Supplementary Figure 4I), supporting the notion that the two hexokinases regulate MML-1 via distinct mechanisms, reflected in the updated model (Figure 5E).